# Event-related functional MRI of awake behaving pigeons at 7T

Mehdi Behroozi [1✉], Xavier Helluy[1,2], Felix Ströckens [1], Meng Gao[1], Roland Pusch [1], Sepideh Tabrik [3], Martin Tegenthoff[3], Tobias Otto[4], Nikolai Axmacher [5], Robert Kumsta[6], Dirk Moser [6], Erhan Genc[1,7] & Onur Güntürkün [1✉]

Animal-fMRI is a powerful method to understand neural mechanisms of cognition, but it remains a major challenge to scan actively participating small animals under low-stress conditions. Here, we present an event-related functional MRI platform in awake pigeons using single-shot RARE fMRI to investigate the neural fundaments for visually-guided decision making. We established a head-fixated Go/NoGo paradigm, which the animals quickly learned under low-stress conditions. The animals were motivated by water reward and behavior was assessed by logging mandibulations during the fMRI experiment with close to zero motion artifacts over hundreds of repeats. To achieve optimal results, we characterized the species-specific hemodynamic response function. As a proof-of-principle, we run a color discrimination task and discovered differential neural networks for Go-, NoGo-, and response execution-phases. Our findings open the door to visualize the neural fundaments of perceptual and cognitive functions in birds—a vertebrate class of which some clades are cognitively on par with primates.

[1] Department of Biopsychology, Faculty of Psychology, Institute of Cognitive Neuroscience, Ruhr University Bochum, Universitätsstraße 150, 44780 Bochum, Germany. [2] Department of Neurophysiology, Faculty of Medicine, Ruhr University Bochum, Universitätsstraße 150, 44780 Bochum, Germany. [3] Department of Neurology, BG-University Hospital Bergmannsheil, Ruhr University Bochum, Bürkle-de-la-Camp-Platz 1, 44789 Bochum, Germany. [4] Department of Cognitive Psychology, Faculty of Psychology, Institute of Cognitive Neuroscience, Ruhr University Bochum, Universitätsstraße 150, 44780 Bochum, Germany. [5] Department of Neuropsychology, Faculty of Psychology, Institute of Cognitive Neuroscience, Ruhr University Bochum, Universitätsstraße 150, 44780 Bochum, Germany. [6] Department of Genetic Psychology, Faculty of Psychology, Ruhr University Bochum, Universitätsstraße 150, 44780 Bochum, Germany. [7] Department of Psychology and Neurosciences, Leibniz Research Centre for Working Environment and Human Factors (IfADo), 44139 Dortmund, Germany. ✉email: mehdi.behroozi@ruhr-uni-bochum.de; onur.guentuerkuen@ruhr-uni-bochum.de

During the past two decades, technological leaps in advanced analysis methods as well as higher temporal and spatial resolution turned functional magnetic resonance imaging (fMRI) into a neuroscientific core technique that is meanwhile also applied to studies in mice and rats[1], rabbits[2], cats[3], dogs[4], song birds[5], pigeons[6,7], monkeys (marmoset[8] and macaque[9]), and even crocodiles[10]. The majority of small animal fMRI studies were conducted under anesthesia. This leaves some room for studies on sensory-driven processes[11–13] and resting-state networks[14], although there is evidence that anesthesia or sedation also affect sensory-driven blood oxygenation level-dependent (BOLD)[15]. Several experiments studied resting-state BOLD fMRI in awake rats and mice[16], but fall short of the requirements of cognition-related studies that involve active participation and decision making. Consequently, most of the few fMRI studies with awake and working animals were conducted with non-human primates[17,18].

While monkeys are highly visually oriented and cognitively capable, they are also expensive and laborious to train. This is radically different with birds, which are also highly visually capable and can be studied with the full spectrum of neuroscientific methods. In addition, the recent findings on avian cognition demonstrated that especially corvids and parrots are cognitively on par with primates[19]. Since even pigeons match macaque performances for short-term memory, abstract numerical competence, and orthographic processing[20], birds represent an excellent animal model for cognition. We therefore decided to establish an active fMRI platform in pigeons. We believe that our approach can easily be adapted to a large number of further bird species and possibly beyond.

We aimed for a procedure that fulfills following criteria: (i) establishment of a swiftly learned behavioral paradigm that keeps stress levels during scanning at minimum, and in which animals can respond, although the head is fixed; (ii) close to zero motion artifacts even during the production of hundreds of bouts of responding; (iii) optimization of a pulse sequence to minimize susceptibility artifacts; (iv) characterization of the species-specific hemodynamic response function (HRF) to achieve optimal modeling; (v) optimization of all relevant parameters to allow for complex event-related fMRI studies with modest trial numbers. To this end, we established a color discrimination fMRI paradigm with awake and behaving pigeons inside the 7T small animal MRI scanner. Our procedure enables us to analyze brain activity patterns during sensory discrimination, response inhibition, and motor execution. Beyond demonstrating a major tool for cognitive neuroscience, we believe that our approach opens up the opportunity to analyze neural activity patterns during diverse tasks in different species of birds. It thus will make it possible to compare mammals and birds, the two most cognitively capable vertebrate classes, in completely novel ways.

## Results

### Pigeons actively discern colors under low-stress levels and minimal motion artifacts.

When fully conscious animals are scanned, stress levels and, concomitantly, motion artifacts can rise. We therefore adapted a protocol with which our pigeons were habituated to a custom-made restrainer (Fig. 1a, Supplementary Fig. 1, and Supplementary Movie 1) for more than an hour. To this end, animals were first implanted under anesthesia with an MRI-compatible pedestal (Supplementary Fig. 1A) that was later used to fix their heads to the restrainer[21]. Then, animals were habituated to this procedure to reduce stress associated with head fixation and scanner noise[7,21]. In brief, the protocol consisted of a 3-week-long habituation procedure of three steps in which birds were first habituated to the experimental environment and the restrainer. In a second step, the animals were habituated to head fixation. Lastly, animals were accustomed to the scanner sound, before finally moving to the actual experiment.

To assess the acute stress response of the animals during the conduct of our habituation protocol, we measured plasma corticosterone (CORT) levels and the heart rate. We verified our stress assessment method by conducting an adrenocorticotropic hormone (ACTH) stimulation test in a separate group of birds. In line with previous studies (e.g., ref. [22]), all birds showed a substantial CORT increase in response to the administration of synthetic ACTH ($n = 3$; mean increase: $64.68 \pm 9.40$ ng/ml after 30 min; Fig. 1b). On the first encounter of a 10-min-long head fixation, we found an increase in CORT with respect to the individual baseline level for each animal tested ($n = 4$; mean increase: $17.26 \pm 4.99$ ng/ml (mean ± SEM); Fig. 1c and Supplementary Table 1). Correspondingly, we found an elevated heart rate in the range of $190 \pm 35$ b.p.m. (Fig. 1d). Both of these parameters were found to be significantly reduced and close to baseline levels after 7 days of habituation ($0.1 \pm 2.9$ ng/ml increase in CORT; heart rate: $150 \pm 13$ b.p.m.; baseline level = $155 \pm 28$ b.p.m.; ref. [23]). A one-way repeated-measures analysis of variance (ANOVA) revealed a significant effect of habituation on stress levels ($n = 4$, CORT concentrations: $F_{(2,6)} = 13.83$, $p = 0.006$; heart rate: $F_{(2,6)} = 27.81$, $p = 0.001$). Follow-up comparisons indicated lower CORT concentrations at habituation day 4 compared to habituation day 1 ($p = 0.016$). Furthermore, the habituation protocol resulted in a significant decrease in heart rate at habituation day 7 compared to habituation day 1 ($p = 0.023$). Thus, we could show that head fixation barely stressed the animals after habituation and moved on to establish an operant response for the conditioning task.

Pigeons minutely open and close their lower jaw to drink small sips of water. We therefore decided to use lower jaw movements (mandibulation) as an operant response and trained our birds to drink from a water receptacle in response to an operant stimulus while being head fixed. Using a pressure-sensitive piezo-electric device mounted under the lower jaw, we could monitor all mandibulations. To avoid interferences between radio-frequency pulses/gradient switches and piezo-electric activity, we added a second piezo-electric sensor as a reference close to the first one to measure mandibulations (Fig. 1a). Head motions were smaller than our voxel size of $0.47 \times 0.47 \times 1$ mm$^3$, even when the animals actively mandibulated (Fig. 1e, f). The maximum absolute translation and rotation during the color discrimination experiment were 0.048 mm and 0.045°, and 0.041 mm and 0.043° during the resting-state fMRI time-series, respectively (Fig. 1e, f and Supplementary Fig. 2). To further evaluate the images with/without frequent mandibulation, we compared the motion parameters for each animal from resting-state scans with low activity with those from phases of frequent mandibulation. Median absolute deviation (MAD) of the estimated motion parameters for the fMRI time-series of the two scans showed no significant differences for translation and rotation (paired-sample $t$ test, $p = 0.18$ (for translation), $p = 0.12$ (for rotation); see Fig. 1e, f). In addition, we calculated the voxelwise temporal signal-to-noise ratio (tSNR) over the resting-state scans and the scans during the color discrimination task (Supplementary Fig. 3). The tSNR of a rapid acquisition with relaxation enhancement (RARE) sequence in each voxel over a 10-min resting-state scan was calculated after high-pass temporal filtering (cutoff at 100 s) and spatial smoothing (full-width at half-maximum (FWHM) = 8 mm, after upscaling voxel size). tSNR in the entire telencephalon ranged from 110 to 220 (Supplementary Fig. 4). The results indicated that mandibulations had no effect on the quality of the images, since tSNR values were highly similar between conditions.

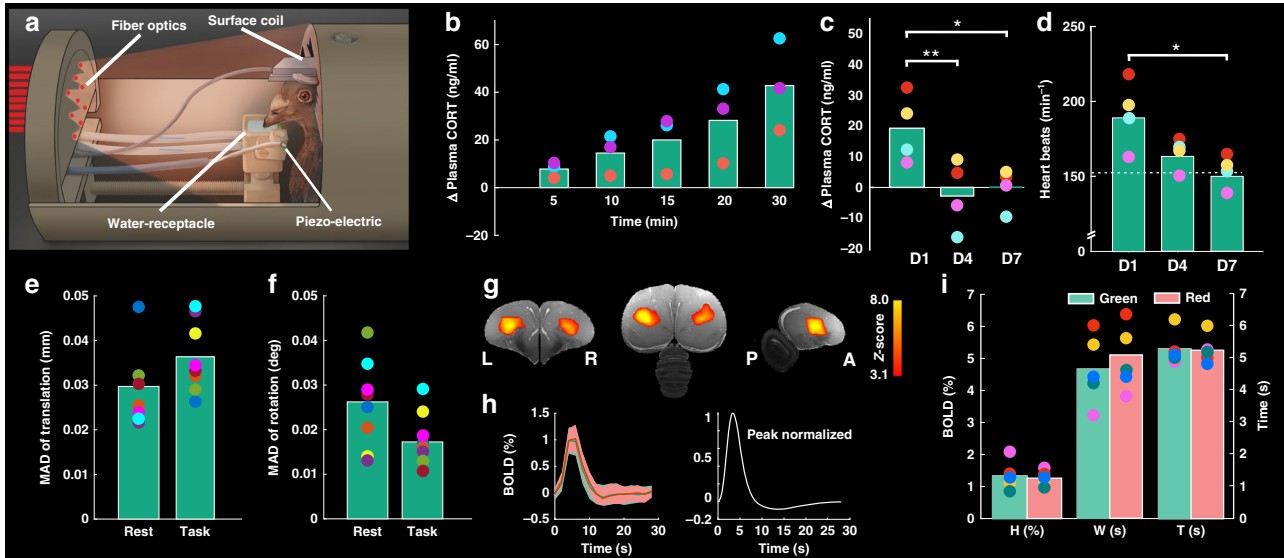

**Fig. 1 Platform for awake pigeon fMRI. a** Head-restrained pigeons were positioned in the scanner bore center and stimuli were presented using fiber optics connected to a light stimulator outside the scanner room. Animal responses were registered using a piezo-electric sensor mounted under the lower jaw. **b** Corticosterone (CORT) increase in response to the adrenocorticotropic hormone (ACTH) stimulation test ($n = 3$). Mean CORT concentration increases over time after ACTH administration. **c** Absolute change in plasma corticosterone on habituation days D1, D4, and D7. Absolute change in CORT was significantly lower on habituation day 4 and day 7 compared to habituation day 1. Bars = mean values of absolute corticosterone level change ($n = 4$). \*\*$p = 0.016$, \*$p = 0.037$ (one-way repeated-measure ANOVA, a Sidak correction for multiple comparison). **d** Heart rate during the habituation procedure on days 1, 4, and 7 ($n = 4$). Heart rate was significantly lower on habituation day 7 compared to habituation day 1 (dotted line = baseline at rest). \*$p = 0.023$ (one-way repeated-measure ANOVA, a Sidak correction for multiple comparison). **e, f** Median absolute deviation (MAD) of estimated motion parameters (translation and rotation) over resting-state and active decision task ($n = 8$). **g** Example of activation clusters caused by 2 s visual stimulation (calculated for an exemplary pigeon brain for the first run) to estimate the hemodynamic response function parameters. The activation mask from individual pigeons was applied to the other runs to extract the time course of hemodynamic responses within each ROI. **h** Left curve shows the BOLD signals (HDR) for different visual stimuli. Data are represented as mean ± SEM ($n = 5$ animals, number of trials per condition is 300). The right curve shows the best-fitted pigeon hemodynamic response function ($\alpha_1 = 7.71$, $\alpha_2 = 11.48$, $\beta_1 = 1.74$, $\beta_2 = 0.74$, $c = 0.25$). **i** Estimated parameters of HDR for different colors. Hemodynamic responses did not differ between colors. Each circle in **b–f** and **i** represent a single value per individual during different experiments. ACTH adrenocorticotropic hormone, CORT corticosterone, BOLD blood-oxygen-level-dependent, H height, T time-to-peak, W full-width at half-maximum, A anterior, P posterior, L left, R right. Source data are provided as a Source Data file.

**The HRF of pigeons differs from that of humans**. To detect neural activity from BOLD fMRI signals, we selected the standard human fMRI analysis software package FSL (https://fsl.fmrib.ox.ac.uk/fsl/fslwiki/FSL, version 5.0.9) and conducted a statistical analysis based on multiple linear regressions. Herein, we used the general linear modeling (GLM) approach. GLM traditionally models the experimentally measured BOLD fMRI response as the convolution of a boxcar stimulation signal with a species-dependent canonical HRF. We therefore experimentally estimated the pigeon HRF based on the analysis of experimental BOLD signals (which are also called hemodynamic responses (HDRs)) from a primary visual region (entopallium). This was done in five pigeons that were shown color stimuli for 2 s in order to model an average HRF that best fits the fMRI time-series over all subjects. Since our birds closed their eyes and sometimes appeared to have fallen asleep, we had to increase their attention to the task (Supplementary Fig. 5). We therefore water deprived the animals before the task and rewarded them with water during stimulus presentation. For each pigeon, we measured five runs with each run, including 15 trials per color. The first run was used to identify regions of interest (ROIs) where brain activation occurred following visual stimulation (Fig. 1g). The next four runs were used to characterize the pigeon HRF. The measured responses from different runs were averaged over five subjects. Data were fitted with a double-gamma distribution function convolved with a 2 s boxcar (Eq. 1 in "Methods"), which resulted in: $\alpha_1 = 7.71$ (95% confidence interval: 6.16–9.23), $\alpha_2 = 11.48$ (95% confidence interval: 2.39–20.57), $\beta_1 = 1.74$ (95% confidence

interval: 1.34–2.14), $\beta_2 = 0.74$ (95% confidence interval: 0.36–1.12), $c = 0.25$ (95% confidence interval: 0.15–0.35) (Fig. 1h). We used this pigeon HRF to process the subsequent fMRI color discrimination task data (Supplementary Methods).

In addition to HRF, we measured height ($H$), time-to-peak ($T$), and FWHM ($W$) of the HDR of the pigeon visual system in response to 2 s visual stimuli (red and green light, Fig. 1i). To check for subtle differences of these HDR parameters, we calculated a repeated-measure ANOVA, with HDR parameters as within-subject factors and light colors as a between-subject factor. The results (Fig. 1i) indicated no significant differences between HDR parameters of the different colors ($F_{(1,8)} = 0.171$, $p = 0.69$). The peak of the BOLD response ($H$ parameter) occurred ~4.5–5.5 s after stimulus onset. The average BOLD response measured in entopallium showed a highest percent signal change at 0.8–1.5%. The width of HDR was in the range of 4–6 s.

**Behavioral responses during the color discrimination task**. We then designed a Go/NoGo color discrimination task (Fig. 2a). Head-restrained and water-deprived pigeons ($n = 8$) were trained to respond to a reward-associated color (S+) and to inhibit their response to a non-rewarded one (S−) (Fig. 2b). Behavioral responses and brain activity were recorded during post-criterion sessions (72 trials per condition, see "Methods"). A jittered inter-trial interval (ITI) paradigm (12.2–20.2 s) was used to ensure unpredictability of the upcoming stimulus and to obtain higher time resolution of BOLD responses. We calculated mandibulation rates (number of responses during stimulus period) during Go

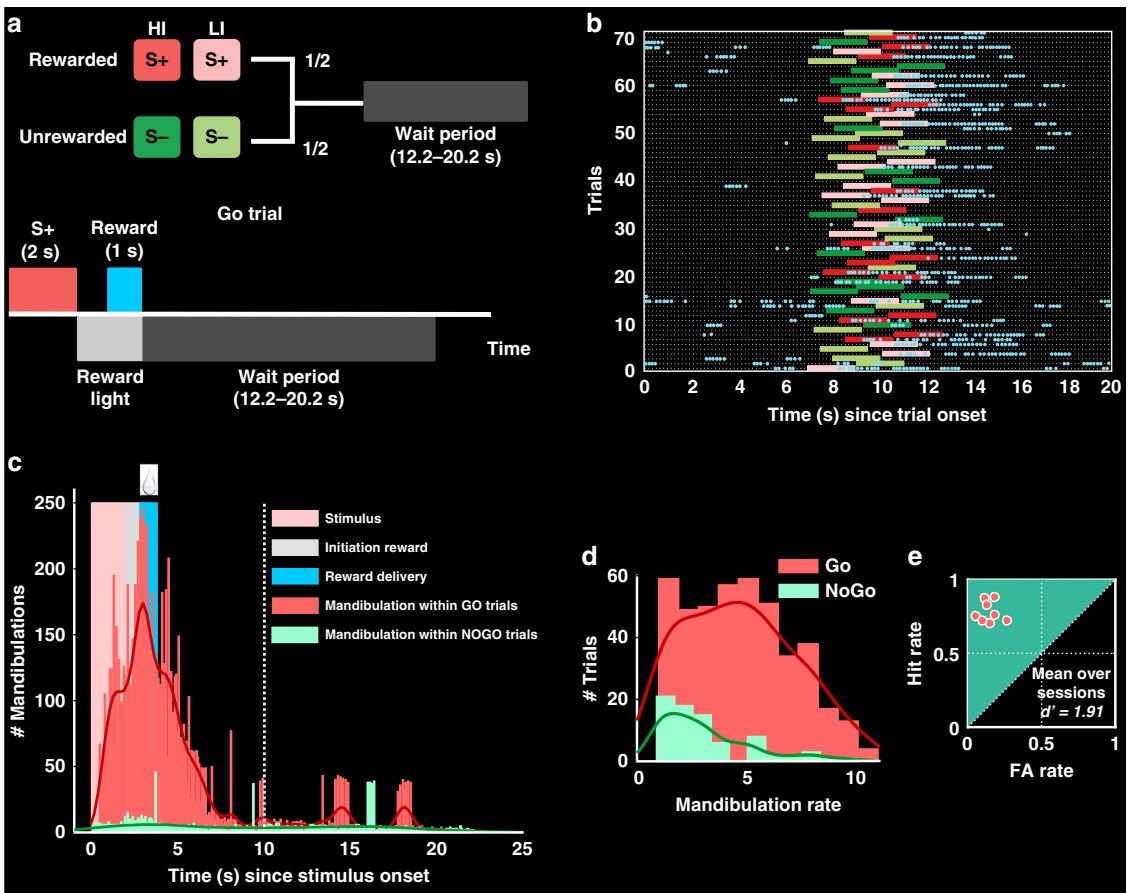

**Fig. 2 Behavioral results of the color discrimination task. a** Top row shows the experimental design. All events were equiprobable. Bottom row shows the timing of an individual Go (S+) trial. Animals were rewarded when mandibulating during S+ presentations. To avoid stimulus intensity effects during training, two different light intensities (20 and 100% of maximum possible intensity) were used. **b** Raster plot of a typical session from one pigeon. Time increase from bottom to top for the session and left to right represents the event within single trials (with time-point 0 corresponding to trial onset). During the depicted session, red-light signaled S+ and green-light S− trials. Mandibulation was tracked during each single trial. Blue dots represent the animal's mandibulations, which happened mostly during S+ presentations and reward periods. **c** Histogram showing an overlay of mandibulations both during Go and NoGo trials in a group analysis of eight pigeons (with time-point 0 corresponding to stimulus onset). Dotted line shows the end of the post-reward period. **d** Mandibulation rate histogram. Group analysis showed that mandibulation rate over all S+ trials was significantly higher than over all S− trials (two-tailed t test, $t(654) = 9.66$, $p = 9.99e − 21$). **e** Behavioral results of all pigeons over all test sessions. Each data point represents the projected Hit/FA pairs to the ROC space. Source data are provided as a Source Data file.

and NoGo periods (Fig. 2c, d) as an estimate of discrimination performance. High hit and low false-alarm (FA) rates across animals represented an optimal trajectory on a receiver operant characteristic (ROC) chart (Fig. 2e). The mean reaction time (first mandibulation after stimulus onset) during S+ and S− trials was $0.9 \pm 0.4$ s (mean ± SD, hits) and $0.7 \pm 0.5$ s (mean ± SD, FAs), respectively.

**Activity patterns during S+ color presentation.** To reveal neural activity during visual discrimination of colors, we used the contrast of Go > NoGo + mandibulation in a first-level analysis. The estimated parameters of brain activation were then entered into a second-level group analysis using random-effect modeling ($Z = 3.1$ and $p < 0.05$ family-wise error (FWE) corrected at the cluster level, group analysis, for more details see "Methods"). Prominent activation clusters were found in many areas of the brain and were outlined in functional groups (Fig. 3). Overall figures are provided in the Supplementary Fig. 6 and below illustrate the network of activations of different subsystems within the pigeon telencephalon.

As expected, many visual areas were active during responses to the S+ stimulus. The avian visual forebrain is constituted by the

visual thalamo and tectofugal pathways[24]. We observed significant activations in the primary thalamic input areas of both the thalamofugal (hyperpallium intercalatum) and the tectofugal system (entopallium). In addition, associative visual areas of the thalamofugal (HA, hyperpallium apicale; hyperpallium dorsale) and the tectofugal system (nidopallium intermediolaterale; nidopallium intermediomediale; nidopallium intermediolatera-teralis; MIVl, mesopallium intermedioventrale, pars lateralis), convergence zones of both pathways (NFL, nidopallium fronto-laterale), as well as higher-order multimodal associative areas (MIVm, mesopallium intermedioventrale, pars medialis; TPO, area temporo-parieto-occipitalis) evinced significant activations (Fig. 3a and Supplementary Figs. 6 and 7).

Within the olfactory system the bulbus olfactorius (BO) as well as target areas of mitral cells were activated (CPP, cortex prepiriformis; TuO, tuberculum olfactorium). In addition, the area surrounding the fibers of the medial olfactory tract (MOT) showed a significant BOLD response. Within the auditory system, all subareas of the field L complex were activated, especially in its most ventromedial part. In addition, the auditory associative nidopallium caudomediale (NCM) and the mesopallium caudale (MC) also showed activity responses. Within the trigeminal and

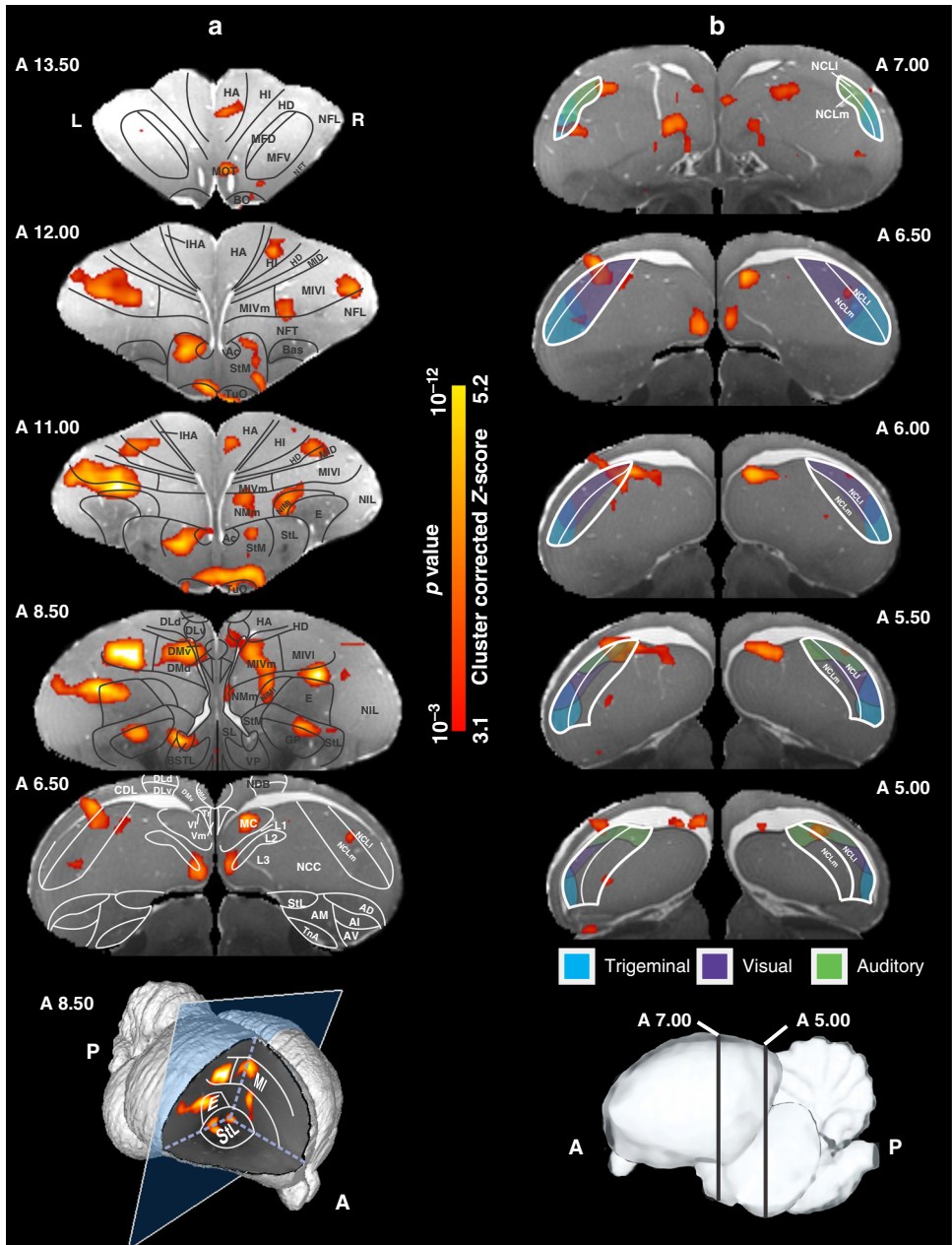

**Fig. 3 BOLD response pattern during the color discrimination task.** Coronal images at different levels of the pigeon telencephalon with overlay of anatomical borders based on the pigeon MRI atlas[73]. Pigeon atlas images are shown in gray scale, while the areas activated significantly during the color discrimination task (contrast of Go > NoGo + mandibulation) are highlighted in color ($Z = 3.1$ and $p < 0.05$ FWE corrected at the cluster level, group analysis). Results demonstrate activated clusters in **a** visual, olfactory, auditory, trigeminal, tactile, and polysensory areas; limbic and basal ganglia components; and **b** memory- and executive control-related pallial areas (see also Fig. 5 and Supplementary Fig. 6). The last image on the left column shows a 3D representation of the pigeon brain with an example window at level A 8.50. Anatomical borders in **b** are based on study by Herold et al.[50] and color-coded areas (red, green, and yellow) represent the terminal fields of afferents from secondary sensory areas on the NCL, based on Kröner and Güntürkün[77]. The last image of the right column shows a 3D representation of the selected coronal images from NCL. HI high intensity, LH low intensity. The corresponding abbreviations of ROIs are listed in the Supplementary Table 3.

the tactile systems, the nidopallium frontotrigeminale (NFT) and the HA evinced significant BOLD responses (Fig. 3a and Supplementary Figs. 6 and 7).

One of the activated limbic structures was the n. accumbens (Ac). This would be expected when a stimulus that signals the prospect of reward occurs. Further active limbic forebrain structures were the area corticoidea dorsolateralis, mesopallium frontodorsale, nidopallium caudocentrale, n. posterioris amygdalopallii pars basalis, the septum laterale, the bed nucleus of the

stria terminalis, pars lateralis, and the n. diagonalis Broca (Fig. 3a and Supplementary Figs. 6 and 7). As would be expected from a highly trained task that includes an operant response, we found significant activations in the lateral and medial striatum, and the globus pallidus, as well as the ventral pallidum. In addition, several subareas of the prefrontal-like and associative nidopallium caudolaterale (NCL) including its lateral and medial subregions were active. A detailed analysis of the location of the BOLD response areas showed that many activity fields were located in

the overlapping zones of the trigeminal, visual, and auditory areas of the NCL (Fig. 3b and Supplementary Figs. 6 and 7).

Dorsal and ventral parts of dorsolateral region, ventral part of the dorsomedial region, triangular region of the ventromedial region, and the ventrolateral part of the V complex (Fig. 3a and Supplementary Figs. 6 and 7) were activated in response to the S+ stimulus. Since hippocampal DM is the major hub of the avian hippocampal system, the activation in this subarea could constitute a gatekeeper function for sensory input to the hippocampus[6].

**Activity patterns during NoGo trials**. In a Go/NoGo task, half of the trials require the animal to not respond when faced with the S−, and in the other half of the trials, animals should respond to the S+, which is associated with reward. To visualize the activity patterns during NoGo trials, we used the contrast NoGo > Go ($Z = 3.1$ and $p < 0.05$ FWE corrected at the cluster level, group analysis). As illustrated in Fig. 4a, we indeed found prominent activation clusters in two areas. One was the lateral habenula (HbL) of the epithalamus (Fig. 4a and Supplementary Fig. 8). In

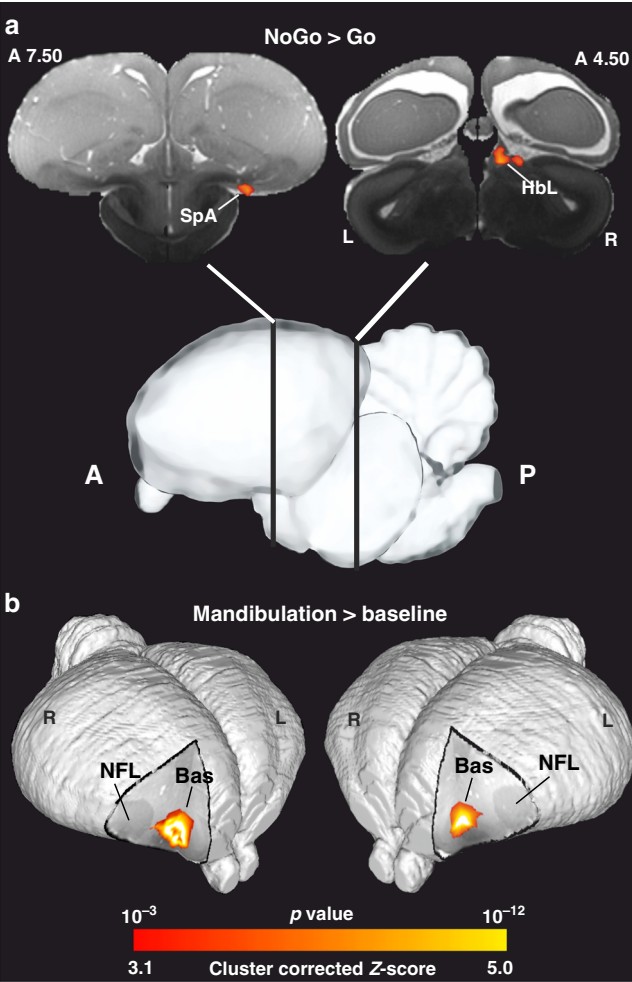

**Fig. 4 Inhibitory and mandibulation response patterns. a** Statistical activation maps for signal increase in the contrast of NoGo > Go ($Z = 3.1$ and $p < 0.05$ FWE corrected at the cluster level, group analysis). **b** 3D representation of statistical activation map for signal increase from the contrast of mandibulation > rest ($Z = 3.1$ and $p < 0.05$ FWE corrected at the cluster level, group analysis). The activation maps were registered and illustrated on the pigeon atlas. The activation significance is demonstrated by the color scale. HbL lateral habenula, SpA subpallialis amygdalae, Bas nucleus basalis, NFL nidopallium frontolaterale.

addition, the area subpallialis amygdalae (SpA) as part of the extended amygdala was found to be active (Fig. 4a and Supplementary Fig. 8). As will be discussed below, these two areas are key components of the vertebrate system for behavioral inhibition.

**The response component**. To disentangle the motor component within the GLM analysis, we modeled mandibulations during resting-state periods and ITI using the filtered piezo-electric signals as an independent explanatory variable. Brain activation in relation to mandibulations (mandibulation > baseline) was found in the nucleus basalis (Fig. 4b and Supplementary Fig. 9), which receives trigeminal input from the beak ($Z = 3.1$ and $p < 0.05$ FWE corrected at the cluster level, group analysis).

## Discussion

We here report the details of a platform that allows running high-resolution fMRI studies in actively responding small animals like pigeons. We conducted a Go/NoGo color discrimination task and uncovered a neural network that could be analyzed with respect to diverse cognitive components. We are confident that our approach will enable advances in systems neuroscience of small birds and possibly beyond.

Cognitive studies with fMRI require awake and active participation–a prerequisite that is difficult to meet when animals must be immobilized. Immobilizing and head fixation increase stress[25] and stress runs contrary to animal welfare requirements, interferes with cognitive processes[26], and can modify HDRs[27]. We therefore improved a stepwise habituation procedure for pigeons[7,21] and checked its progress by measuring heart rate and plasma CORT levels as acute stress indicators. In accordance with results of previous studies[25], all our birds showed an increase in circulating CORT on the first day of habituation. Subsequently, plasma CORT significantly decreased to individual baseline levels over the course of habituation. Similar to the results of Cockrem and Silverin[28], our data indicate a habituation to the applied stressors of capture, handling, and restraining. Correspondingly, the heart rate decreased to the baseline level of ~150 b.p.m.[23] during habituation. Taken together, these values indicate that the animals experienced low levels of stress once they were fully habituated to the experimental procedures. Concomitant with stress reduction, also motion artifacts were reduced to ~0.05 mm during mandibulation and were thus lower than in similar studies with birds under conditions of passive sensory stimulation[7,13]. Thus, our platform enables fMRI studies in actively participating small birds with high motion artifact control under low-stress conditions.

To achieve whole-brain coverage during scanning, we had to refrain from using a conventional gradient-echo sequence at 7T used for birds in the past, since it restricts the field of view (FOV) due to strong local magnetic field inhomogeneities caused by air cavities in the bird skull[5,12]. We instead worked with a RARE sequence. A 4X segmented RARE sequence has been used earlier to map responses to auditory[12] and visual stimuli in birds[7] with a repetition time (TR) of 8 s and an echo time (TE) of 60 ms. However, this only allows running block design experiments and does not provide sufficient sensitivity and temporal resolution for event-related designs. In the current study, we thus used the single-shot RARE sequence adopted from Behroozi et al.[6] for our experiments, which overcomes these limitations and permits recording whole-brain responses with an in-plane resolution of $0.47 \times 0.47$ mm$^2$ within <4 s during an active discrimination task. Although the TR could have been kept at 2 s for the 11 slices covering the whole brain as in Behroozi et al.[6], we refrained from doing so to further reduce animals' stress levels by reducing

gradient noise and heat caused by radio-frequency pulses. In addition, we optimized BOLD sensitivity of this RARE sequence by adjusting the effective TE ($TE_{eff}$) to 42 ms, a value close to the typical T2 relaxation times of pigeon brain tissue[21]. Since pigeons possess relatively large eyes and moving them can induce artifacts in the brain regions close to them, we used saturation slices to saturate eye signals, which helped to remove some image artifacts. However, we note that, compared to BOLD fMRI EPI-based sequences, the RARE sequence is much less sensitive to magnetic field fluctuations created by breathing or mandibulations and does suffer only little from geometrical distortions. These changes enabled our event-related color discrimination study in pigeons.

Birds surpass humans with respect to color vision and are easy to condition in operant tasks when having to discriminate between hues[29]. Under conditions of immobilization, however, the choice of the operant response becomes a problem. We chose mandibulation since this a biologically significant behavior during drinking. Indeed, our subjects readily learned this association and quickly reached high discrimination performances with abundant response rates.

Usually, the human canonical HRF is used to estimate activation patterns in different non-human animals, ranging from primates to birds[13,17]. Since avian physiology differs in several key parameters from mammals, such a procedure could produce wrong parameter estimations during bird fMRI[30]. We therefore characterized the pigeon HRF from BOLD responses to visual stimulation in the primary visual entopallium and used it as a canonical pigeon HRF for all the other GLM analysis. While humans and pigeons do not differ with respect to the width of their positive BOLD, time to peak was ~2 s faster in pigeons. It was also faster than in the visual cortex of macaques[9,31] and much faster than in the primary visual area in Nile crocodiles[10]. In terms of signal strength, the maximum BOLD signal change in pigeons was lower than that in comparable primate studies (macaques: ref. [31]; humans: ref. [32], a difference that is likely due to the overall lower sensitivity of the multiple spin-echo sequence used relative to the gradient-echo sequences).

Taken together, our platform enables event-related fMRI studies with high-resolution and whole-brain coverage in actively participating small birds under conditions of very low motion artifacts and low-stress conditions. Up to now, these conditions could only be achieved in primates[17,18]. The only exception of which we are aware is a recent behavioral fMRI study that describes a behavior platform in which mice discriminate odors using a block design[33]. Unfortunately, the authors report a very high level of motion artifacts and so had to discard 21% of the sessions. Susceptibility mismatches were reduced by filling the middle ear with a low viscosity silicone sealant—a procedure that would run contrary to animal welfare obligations of the European Union. In addition, due to low SNR, Z-score activation maps had values lower than the minimum value required to find significantly activated clusters at the whole-brain level[34]. All this unfortunately severely limits the application of this approach. In summary, we are inclined to believe that our procedure represents a much more feasible approach for cognitive fMRI studies in birds and possibly further species. In particular, the proposed platform should scale well to smaller bird species (like the zebra finch). Assuming conservatively a maximum motion displacement of 0.05 mm and directly adapting the described single-shot RARE protocol to the FOV of the zebra finch brain (16 × 16 mm²)[13] would result in a imaging in-plane resolution of 0.25 × 0.25 mm² with motion displacements smaller than a fifth of the voxel size. These numbers should allow a broad range of fMRI investigations.

We visualized the avian neural network for an appetitive Go/NoGo color discrimination task with a time resolution of 4 s.

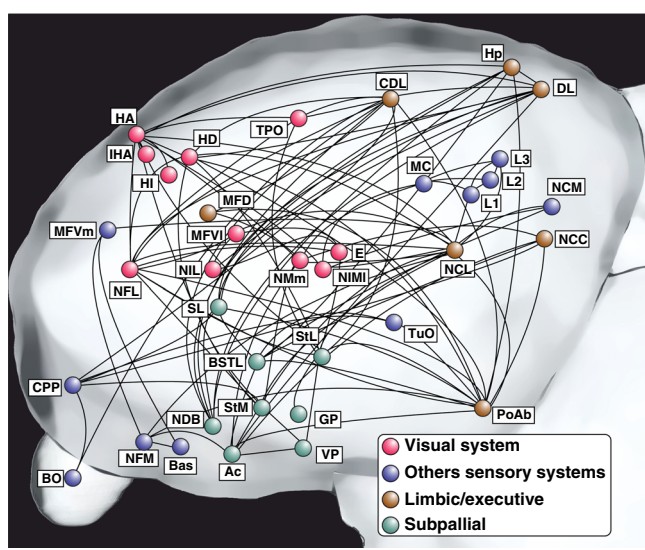

**Fig. 5 Schematic depiction of the telencephalic network in the pigeon brain active during our color discrimination experiment.** Nodes represent active regions in the "Go > NoGo + mandibulation" contrast (for more details see Fig. 3 and Supplementary Fig. 6). Nodes are colored based on their involvement in different functional networks. Note that the activated networks are spatially distributed rather than localized. For abbreviations see Supplementary Table 3.

Thus, the activity of the contrast "S+ color presentation" (Go > NoGo + mandibulation) involves the sum of stimulus encoding, stimulus–response–outcome association, decision making, and reward consumption (Fig. 5). Since our pigeons had to base their choices on stimulus color, we expected to reveal BOLD responses in visual forebrain areas. The avian pallium is reached by both the differentially specialized tecto- and the thalamofugal pathways, which are homolog to the mammalian extrageniculocortical and geniculocortical systems, respectively[35,36]. We found activations of the primary visual entopallium and a large number of visual associative areas (TPO, MIVl, TPO, NFL) that are known to be involved in visual processing[37,38]. Also, the associative visual HA was active—an area with descending projections to sub-telencephalic visual and motor areas[39,40].

In our task, the delivery of the water reward was also associated with multiple non-visual cues like the soft sound of the water pipe valve and the smell of the inflowing water, as well as the trigeminal and tactile feedback during water consumption. It has been shown that such activities in early sensory cortices of primates reflect perceptual experience, rather than the presence of an external sensory stimulation[41,42]. A study by Meyer et al.[41] demonstrated that visual stimuli, which are associated with auditory stimuli, can activate the auditory system in human brain even in the absence of the auditory stimulus. In addition, Zhou and Fuster[42] could show that during a visual–haptic associative learning task in monkey, somatosensory neurons would also respond to visual stimuli, when the visual stimulus has been associated with a haptic stimulus before. In line with these results, we found activity in multiple primary and associative auditory (field L1–3, NCM, and MC)[38,43] and olfactory (BO, CPP, MOT, and TuO)[44,45] areas, likely caused by an association of auditory and olfactory stimuli to the reward. We also identified BOLD responses in the trigeminal and tactile pallial areas (HA and NFT)[46] that possibly result from somatosensory feedback during reward ingestion. In addition, the polysensory MIVm that receives input from various sensory and limbic structures was active[38]. Thus, the post-choice presence of multiple reward-associated cues created multisensory activations at the forebrain

level. To prevent such activations, future studies could make use of a symmetrically rewarded Go/NoGo design[47], which balances reward expectation and association components, which could thus be removed from the analysis (see Supplementary Note 1 for more details).

Our pigeons worked for reward. Consequently, the "S+ color presentation" contrast enabled us to visualize activations of key reward areas. Foremost, this was the dopaminergic ventral tegmental area (VTA) and its projection area n. Ac[48,49]. In addition, further pallial areas that receive dopaminergic input were strongly activated, such as amygdala, hippocampus, striatum, and pallidum[50]. We also observed BOLD responses in the NCL, the avian functional analog to the prefrontal cortex[51]. The dopaminergic input to NCL[51] activates D1 receptors of principal neurons[52]. In pigeons, D1A and D1D receptors mediate learning of stimulus–response associations[53]. It is likely that the activation of these receptors provides a prediction error signal to NCL neurons that enables them to code for stimulus reward value, upcoming choice, and action outcome[54]. All of these are cognitive components of the Go/NoGo procedure.

A Go/NoGo task requires the animal to refrain from responding in S− trials. Studies in different vertebrates show that neurons of the HbL code for negative events[55] and are able to downregulate the activity of dopaminergic VTA cells[56]. As a result, the HbL plays a key role in aversion-induced behavioral suppression by generating a feedback signals that discourages behavioral repetition[57]. Indeed, we discovered a strong HbL activation during NoGo trials. But our NoGo trials were also associated with a BOLD response in the SpA area, which is part of the avian central extended amygdala[58]. Studies in mammals show that this area orchestrates behavioral inhibition via its subcortical projections[59]. Thus, the activation of HbL and SpA during NoGo trials could go along with a decrease of dopaminergic transmission and a concomitant overall activity reduction.

The reverse happens at S+ onset when pigeons start to respond. The contrast "mandibulation > baseline" evinced a prominent activation cluster in the pallial Bas region of pigeons. This nucleus receives trigeminal input and provides an indirect projection to the motor arcopallium, which controls beak movements during ingestive behavior[60,61].

Taken together, we established an avian fMRI protocol that opens the door to analyze awake neural network functions in actively discriminating pigeons without using anesthesia. Our procedure allows robust event-based fMRI experiments in Go/NoGo designs with TRs varying between 2 and 4 s. Obviously, we see the color discrimination of the present paper just as a proof of principle for more complex cognitive studies. Thus, our platform can be used to visualize 3D activity patterns during cognitive tasks in a vertebrate class of which some clades are cognitively on par with primates.

## Methods

**Animal preparation**. All experimental procedures were conducted in strict accordance with the stipulated guidelines for care and use of animals provided by an ethics committee of the State of North Rhine-Westphalia, Germany (Landesamt für Natur, Umwelt und Verbraucherschutz Nordrhein-Westfalen (LANUV), Application number: Az.:84–02.04.2014.A206). Twenty adult Columba livia pigeons (weight: 150–200 g, 3 years old, undetermined sex), obtained from local breeders, were used for all experiments (see Supplementary Table 2 for more details). All animals were individually housed in wire-mesh cages with a 12 h light on/off cycle and provided with water and food ad libitum during the recovery period. During the experimental phase, animals were water deprived for the whole night before the experiment to increase their motivation to participate during the task (see below). After finishing an experiment, the pigeon had free access to water until the next deprivation round started.

Prior to MR experiments, each pigeon was implanted with an MR-compatible plastic pedestal (Supplementary Fig. 1A) attached to the animals' skull. All operations were performed under ketamine/xylazine (70% ketamine, 30% xylazine, 0.075 ml/100 g) supplemented by gas anesthesia (Isoflurane; Forane 100% (V/V),

Mark 5, Medical Developments International, Abbott GmbH and Co. KG, Wiesbaden, Germany). For surgery, the animals' heads were fixed in a stereotactic apparatus[62]. After conducting a skin incision, four holes were drilled into the anterior and posterior surface of the skull. Polyether ether ketone (PEEK) micro pan head screws were inserted into the holes in order to attach a custom-made plastic pedestal. Then, pedestal and screws were embedded in dental cement (OmniCeram) to increase the adhesive strength between skull and pedestal. Following each surgery, analgesic (Carprofen (Rimadyl), 10 mg/kg) and antibiotic (Baytril, 2.5 mg/kg) treatment was given every 12 h for at least 3 days. After a recovery period of 6–8 weeks in which animals had free access to water and food, the habituation training started.

**Habituation for MRI scanning under awake condition**. To reduce the stress induced by the head fixation and to minimize body motion artifacts during the experiments, all animals were habituated to the holding device (Fig. 1 and Supplementary Fig. 1; this device was used both as a mock scanner during the habituation protocol and as the holding device during scanning) and to the scanner noise prior to the behavioral conditioning and prior to the awake scanning. The used habituation protocol is a well-established procedure in our laboratory[6,7,21] for short fixation periods, but validated in this study for longer periods of animal fixation above an hour. In brief, the protocol consisted of three consecutive steps: (i) to habituate the animals to the experimental environment and the restrainer, pigeons were wrapped in a cloth jacket and placed in the restrainer. The restrainer was then moved into a mock scanner setup located in a dark room to reduce context-related stress and excitability; (ii) to habituate the animals to the head fixation, pigeons were fixed via the implanted plastic pedestal on their skull to the restrainer for increasing time intervals starting from 10 to 100 min; (iii) to habituate pigeons to the scanner noise, a recording of the real scanner noise was played to the animals. Sound pressure level was slowly increased until it reached 90 dB, which represents the actual noise level in the scanner room. After an average of ~20 days of habituation, pigeons showed no recognizable stress responses. This was the starting point of the actual experiment.

**Estimation of the acute stress response**. Head fixation ensures a minimal amount of motion of the brain during scanning. It has been shown to be especially effective in bird species due to their specific bone structure[63] and lack of mastication muscles[64]. However, to avoid stress-related artifacts on the BOLD signal[27] and any unanticipated movements that could induce artifacts or would lead in worst case to the detachment of the pedestal, pigeons need to be accustomed to all experimental procedures. Thus, it is mandatory for both scientific and for animal welfare reasons, to implement a habituation protocol and to ensure its effectiveness. Therefore, we acquired two well-established and reliable indicators of acute stress: We monitored the heart rate during the habituation procedure and measured the plasma CORT level (e.g., ref. [25]).

**Blood sampling procedure and plasma preparation**. All blood samples were collected from the brachial or metatarsal vein. The puncture sites were disinfected with 70% ethanol and locally anesthetized with Xylocaine (10%, AstraZeneca, Wedel, Germany). The blood was transferred to ethylenediaminetetraacetic acid-coated tubes (Sarstedt, Microvette CB300 K2E) to prevent blood coagulation. The tubes were immediately centrifuged at 4 °C (Hermle Z216MK, 1900 r.c.f.) and the supernatant plasma was collected and stored in small tubes at −80 °C until further analysis. The ACTH stimulation experiments (see below) were performed in a successive order on one day during the afternoon (13:30–17:45) since birds exhibit a diurnal rhythm of CORT concentrations[65]. During the procedure, pigeons were restrained to prevent additional stress due to successive catching. For stress assessment during habituation, we performed measurements on days 1, 4, and 7 of the habituation procedures. Again, experiments were performed in a successive order on each day during afternoon (13:30–17:45).

**Plasma analysis**. The plasma concentration of CORT was quantified using an enzyme-linked immunosorbent assay (ELISA), based on the principle of competitive binding (CORT rat/mouse ELISA DEV9922; Demeditec Diagnostics GmbH, Kiel, Germany). In these assays, different concentrations of CORT in a sample lead to systematic changes in the optical density of a coated microplate well. The assay was performed following the guidelines provided by the manufacturer. The absorbance, and thus the respective concentration of the CORT, was read out using a Microplate Reader (BioTek, Synergy 2, Winooski, VT, USA) at two different wavelengths of 450 and 630 nm. Samples were always analyzed in duplicates or triplicates.

**ACTH stimulation test**. To verify our stress assessment, we first performed an ACTH stimulation test. This test is routinely performed to assess the functioning of the adrenal glands stress response[22]. ACTH is a hormone produced by the anterior pituitary gland that stimulates the adrenal glands to release CORT, the major avian glucocorticoid[66]. For the test, three birds received intravenous (i.v.) injections of 125 μg of tetracosactid, a synthetic form of ACTH (Synacthen, Novartis Pharma GmbH, Nürnberg, Germany). For each individual bird, an initial sample of blood served as baseline and was taken immediately before the drug injection. Successive

blood samples were taken 5, 10, 15, 20, 30, 50, and 60 min after drug application, to investigate the resulting CORT release. We found an increase of CORT concentration over time. A comparison to a previous study[22] confirmed our method to measure CORT concentrations in plasma samples of pigeon blood. Importantly, an increase in the stress response based on CORT level determination could reliably be measured even after 10 min (Fig. 1c, d).

**Stress assessment during the habituation procedure**. For the assessment of acute stress during the habituation procedure, we measured the circulating CORT concentrations immediately before a habituation session started. Ten minutes after the habituation procedure, a second blood sample was collected, and the circulating CORT concentrations were assessed. Four animals participated in these experiments. It has been shown that there is marked variation between individual birds in their baseline CORT concentrations and their stress-induced CORT responses, albeit individual responses are generally repeatable[28,67]. To cope with inter-individual variability in CORT baseline values (mean: 100.08 ± 79.7 ng/ml), the initial blood sample of each bird served as individual baseline. The change in circulating CORT for each individual bird was calculated and values are expressed as absolute CORT changes[68] (see Fig. 2c). In addition, the heart rate of these animals was recorded during the habituation process (Lutech V7 Veterinary monitor, Ronkonkoma, NY, USA).

**Experimental setup**. For the experiment, animals were placed in the restraining mock scanner with the head being fixed via the implanted pedestal to the restrainer. Animals' behavior was monitored carefully during all training phases by observation or camera to ensure that they could open their lower jaw (mandibulation) as the operant response without any obstruction. Mandibulations were registered by a piezo element positioned below the lower jaw of the pigeons (Fig. 1), with the piezo-electric pressure sensors measuring the dynamic pressure of jaw movements. To avoid interference between radio-frequency pulses/gradient switching and the piezo-electric element, we added a second piezo-electric element as a reference close to the first one. The differential piezo-electric signals served to robustly measure mandibulation. To reward the animals during the task, a water receptacle was positioned at the beak tip of the pigeon, which was filled with 0.2 ml of water for 1 s when the animal gave a correct response (see below). The delivery and removal of the water reward was achieved by using two peristaltic pumps (SR 10_50 12 V DC Tubing Novoprene). A digital input/output interface (Arduino Mega 2560) was used to control all devices used in the training procedure. Stimuli were generated using a 3 × 3 LED matrix (TLC5947 PWM LED Driver) and were presented to the animals inside the scanner via fiber optics (diameter of 2 mm). All computer programing for the experiment was done using MATLAB (2016b, MathWorks, USA) and programs are available on the authors' webpage.

**Visual stimulation paradigm**. Five pigeons (see Supplementary Table 2) were stimulated with visual stimuli. These stimuli consisted of a brief (2 s) green or red-light stimulus, which was presented binocularly in a pseudo-random order. Intensity and wavelength of the different light stimuli were measured in a dark room by a spectrometer (Flame spectrometer, Ocean Optics, Germany) placed 1 cm in front of the optical fiber. The dominant wavelength for red/green-light stimuli was 630 or 517 nm, respectively, while intensity of stimuli was 28.9 cd/m$^2$ for both colors.

The whole experiment consisted of five runs with each run containing 30 trials (15 trials per stimulus). Each trial contained 2 s of stimulation followed by 28 s of darkness to ensure full recovery to baseline. In each run, 490 whole-brain volumes were recorded. Thirty volumes were recorded before the experimental stimulation from which the first 15 volumes were discarded to compensate for T1 saturation artifacts. Four hundred and fifty volumes were recorded in parallel to the experimental stimulation and the remaining 10 volumes after the stimulation. Since animals were rather relaxed during this experiment and tended to fall asleep in the dark scanner environment, we had to increase their attention to the task. To do so, the animals were water deprived for an entire night before the visual stimulation experiment. During the task, animals were supplied with water during the stimulation period. Drinking-related mandibulations in response to water delivery were tracked using the piezo element.

**Color discrimination paradigm**. Eight pigeons (see Supplementary Table 2) were trained in a color discrimination paradigm. Water-deprived animals learned to respond to a reward-associated color by moving their lower jaw and to inhibit their response to a non-rewarded color. Each correct response was rewarded with 0.2 ml water (Fig. 1 and Supplementary Movie 1). To rule out a reward anticipation effect on the BOLD signal induced by the color of the stimulus, animals were divided into two groups (four pigeons per group). For one group, the red light served as the S+ (Go, reward coupled) and the green light as the S− (NoGo, not reward coupled, inhibition of mandibulation required), while for the other group the opposite was employed. In addition, to ensure that pigeons were attending to the light color but not to intensity, two different light intensities for both colors (20% or 100% of the maximum light intensity 28.9 cd/m$^2$, respectively) were used during the experiment. As our results show, animals were indeed responding to color but not to light

intensity (paired-sample $t$ test, $p = 0.69$ (Go trials), $p = 0.62$ (for NoGo trials) (Supplementary Fig. 10).

**Autoshaping and basic conditioning phase**. After habituation to the mock scanner (see above), animals were trained for the color discrimination task with an operant conditioning paradigm. The whole training procedure consisted of an autoshaping and an acquisition phase. In the first step of autoshaping[69], we adopted a classical conditioning training procedure. One training session on each training day consisted of 30 trials. In each trial, only the CS+ (red/green light as conditioned stimuli for the corresponding groups) was presented for 2 s and followed by a water reward (unconditioned stimulus) which was also signaled by two white lights delivered through two additional fiber optics (reward light). During this phase, the CS+ was rewarded irrespective of the mandibulation response (unconditioned response) of the animals. After reward, a randomly jittered 12.2–20.2 s waiting period was used as the ITI. Pigeons progressed to the next training stage when they mandibulated in response to water delivery in at least 80% of the trials on 2–3 consecutive training days.

During the second stage of autoshaping, the training procedure shifted to an instrumental conditioning procedure where the animals received the water reward only when they responded to the S+ during the stimulus presentation period. We denominate this stimulus as S+ since it represents the first-order instrumentally conditioned stimulus that, upon mandibulation as a conditioned response, is followed by reward. The water reward was delivered immediately after the animal mandibulated. There were 30 trials in each training session and the ITI was used as in the previous stage. Immediately after pigeons achieved the performance criterion of 70% correct responses on 1–2 consecutive training days, they entered the third stage of conditioning.

In this third stage, the reward delivery was postponed to shortly after the end of stimulus presentation to avoid a conflict between reward and stimulus. Thus, animals were only rewarded 800 ms after the S+ offset. During this time, a white-colored light served as a reward signal when the animal had mandibulated in response to S+.

During all conditioning phases, pigeons were fixated inside the scanner and a "dummy" fMRI experiment was run. In this study, animals were trained inside a real scanner, but the training phase also could be done outside the scanner environment inside a mock scanner. Animals entered the main experiment when they mandibulated correctly in at least 80% of the trials on 1–2 consecutive training days.

**Acquisition phase**. The main experiment procedure consisted of two equiprobable trial types (S+ and S−; Fig. 2a). S+ and S− trials were presented in a pseudo-random order. S− trials were identical to S+ trials, with the exception that animals had to withhold the response (mandibulation) during S− presentation and were not rewarded. Progress during the main task was monitored using signal detection theory (SDT) measures. S+ trials qualified as hit when pigeons mandibulated during the Go period. If there was no mandibulation during S+ presentation, it was considered a miss. S− trials, in which animals did not mandibulate during stimulus presentation were designated as correct rejection (CR). Mandibulations during S− presentation were considered FAs. As soon as animals showed a performance of at least 80% correct responses in this stage on three consecutive sessions, they entered the final testing phase which was identically designed. In this final phase, each animal performed two sessions in one run. Each run consisted of 10 min for an initial resting-state scan, a first session of 72 trials, 10 min for an intermediate resting-state scan, a second session of 72 trials, and 10 min for a final resting-state scan. The total experiment took 78 min.

**MRI and fMRI procedures**. All MRI experiments were carried out in a Bruker BioSpec small animal MRI scanner system (7T horizontal bore, 70/30 USR, Germany). An 80 mm transmit quadrature birdcage resonator was used for radio-frequency transmission and a single-loop 20-mm surface coil (Supplementary Fig. 1B) for signal detection. The ring surface coil was positioned around the head to reduce motion artifacts resulting from body movements. The Bruker ParaVision 5.1 software was used to acquire the MR data. During MRI runs, respiration waveform was measured using a small pneumatic pillow placed under the pigeon's chest muscles (Small Animal Instruments, Inc. Model 1025T monitoring and gating system).

At the beginning of each imaging session, scout images were measured to localize the position of the animal's brain. Three runs (horizontal, coronal, and sagittal) were acquired using multi slice rapid acquisition (RARE) with the following parameters: TR = 4 s, TE$_{eff}$ = 40.37 ms, RARE factor = 8, no average, acquisition matrix = 256 × 128, FOV = 32 × 32 mm, spatial resolution = 0.125 × 0.25 mm$^2$, slice thickness = 1 mm, number of slices = 20 horizontal, 17 sagittal, and 15 coronal. Based on these images, 11 coronal slices (three slices for auditory and visual paradigms) with no gap between slices were oriented in a way (~45° with regard to coronal direction) to cover the entire telencephalon. These slices were acquired using a single-shot multi-slice RARE sequence adapted from Behroozi et al.[6], with the following parameters: TR = 4000 ms (TR = 2000 ms for visual paradigms, which were used to characterize pigeon HRF), TE$_{eff}$ = 41.58 ms (the TE$_{eff}$ was chosen such as to match T2 of pigeon brain[21]), partial Fourier transform

accelerator = 1.53, encoding matrix = 64 × 42, acquisition matrix = 64 × 64, FOV = 30 × 30 mm², in-plane spatial resolution = 0.47 × 0.47 mm², radio-frequency pulse flip angles for excitation and refocusing = 90°/180°, slice thickness = 1 mm, no slice distance, slice order = interleaved, excitation and refocusing pulse form = scanner vendor gauss512, receiver band width = 50,000 Hz. Two saturation slices were positioned manually to saturate most of the signal from the eyes to avoid brain image corruption due to eye movements.

Each run of the color discrimination task included 1170 (150 volumes per initial, medial, and final resting state) volumes with the first 10 volumes being discarded to compensate for T1 saturation artifacts.

In-plane anatomical images with the same orientation and location as the functional slices were acquired using a RARE sequence with the following parameters: TR = 4000 ms, TE = 36.52 ms, RARE factor = 8, FOV = 30 × 30 mm², acquisition matrix = 128 × 128, in-plane spatial resolution = 0.23 × 0.23 mm², number of slices = 34, slice thickness = 0.25 mm. Images were later used to increase the accuracy of registration of individual functional data to the high-resolution anatomical images. The total scan time was 4 min.

For spatial normalization, high-resolution T2-weighted images were acquired using a RARE sequence with the following parameters: TR = 2000 ms, $TE_{eff}$ = 50.72 ms, RARE factor = 16, number of averages = 1, FOV = 25 × 25 × 15 mm³, matrix size = 128 × 128 × 64, spatial resolution = 0.2 × 0.2 × 0.23 mm³. Total scanning time was 17 min.

**Behavioral data analysis.** All data analysis was done in MATLAB (2016b, MathWorks, Natick, MA USA) using custom-made scripts. The SDT sensitivity index ($d'$) was used to evaluate the behavioral performance. Reaction time was measured as the time between the stimulus onset to the first mandibulation. Mandibulation rate was measured based on the number of mandibulations within S + stimulus windows (2 s).

**fMRI data processing for the color discrimination paradigm.** All fMRI data analysis was performed using tools implanted in the FMRIB Software Library (https://fsl.fmrib.ox.ac.uk/fsl/fslwiki/FSL, version 5.0.9[70], Supplementary Methods) and custom-made scripts in MATLAB. Before starting preprocessing, we had to take care of artifacts from eye movements. The eyes of pigeons are about the same size as the brain. Since eyes moved during the experiment, we segmented out parts of the fMRI images containing eye positions from the 4D datasets, prior to estimating the image motion parameters. The following standard data preprocessing steps were applied: (i) upscaling the voxel size by a factor of 10, (ii) motion correction using MCFLIRT (FSL's intra-modal motion correction tool)[71], (iii) slice time correction (interleaved acquisitions), (iv) removal of non-brain tissue from functional data (using a brain mask), (v) spatial smoothing (using a Gaussian kernel of FWHM = 8 mm, after upscaling), (vi) global intensity normalization by a single multiplicative factor for each scan run (for group analysis), (vii) high-pass temporal filtering (with cutoff at 60 s), and (viii) registration of the color discrimination data to the high-resolution anatomical images. To improve co-registration, functional data were first aligned to the in-plane anatomical images using affine linear registration (six degrees of freedom). Then, in-plane anatomical images were co-registered to the high-resolution T2-weighted RARE anatomical images using affine linear registration (12 degrees of freedom). A population-based template was generated based on the mean 3D high-resolution anatomical images of all subjects (using the FLIRT function in FSL). After analyzing individual subjects, the results were normalized to the low-resolution population-wide template for group analysis. The study by Mueller et al.[72] demonstrated that with higher resampling resolutions, the FWE-corrected $p$ values decrease systematically so that more and more false positives occur. Since the pigeon atlas provides high-resolution images (0.078 mm³), a group analysis (including thresholding) was done in low-resolution population-based space to avoid false positives. For visualizing the results, the group results were co-registered linearly (trilinear interpretation) to the pigeon MRI atlas[73]. 3D MRI images were visualized using the Mango software (http://ric.uthscsa.edu/mango/, version 4.1).

**fMRI data processing for characterizing the HRF.** Data of the first run was used as a brain activation localizer to identify ROIs in which changes in brain activity followed visual stimulation. The image preprocessing steps were the same as above. A GLM analysis was needed only to process the data from the first run in order to find light-activated brain regions. As heuristic, the explanatory variables were convolved with the human canonical HRF using default parameters (double-gamma function, $\alpha_1$ = 6, $\alpha_2$ = 16, $\beta_1$ = $\beta_2$ = 1, $c$ = 1/6). For the visual experiment, explanatory variables were red or green light with a duration of 2 s. The activation maps were calculated based on contrast maps for all light stimuli (red + green light) versus rest. The activated ROIs where then used to extract the BOLD signal time-series from the next four runs. The time-series within each ROI were averaged over the respective voxels. To characterize the baseline signal level, the polynomial regressors were projected out from the mean BOLD signal time-series of each run[74]. BOLD signals were converted to the percentage signal change by subtracting and dividing each time-series by the average of 2 s prior to stimulus onset. To model species-specific HRF, the signal from the selected ROI was fitted with a double-gamma distribution function convolved with a 2-s boxcar (Eq. (1));

representing 2 s stimulus duration) using a nonlinear least-square fit (curve-fitting toolbox in MATLAB, 2016b, the MathWorks, Natick, MA USA). The fitted parameters for the double-gamma function[75] were as follows:

$$y(t) = \left[ \prod(t,0) - \prod(t,2) \right] \otimes \left[ A \left( \frac{(t)^{\alpha_1-1}\beta_1^{\alpha_1}e^{-\beta_1(t)}}{\Gamma(\alpha_1)} - c\frac{(t)^{\alpha_2-1}\beta_2^{\alpha_2}e^{-\beta_2(t)}}{\Gamma(\alpha_1)} \right) \prod(t,0) \right],$$
(1)

where $\prod(t,t_0) = 1$ if $t \geq t_0$ or $t = 0$ if $t < t_0$, $\otimes$ represents convolution, $t$ refers to time, $A$ indicates the amplitude, $\alpha$ and $\beta$ indicate the shape and scale, respectively, $c$ determines the ratio of the response to undershoot, and $\Gamma$ represents the gamma distribution function. To minimize the risk of local-minima solutions, we determined the plausible range for fit function parameters based on the pilot study.

**Statistics and reproducibility.** Whole-brain statistical analysis was carried out using FEAT (FMRI Expert Analysis Tool), a part of FSL. The GLM contained the following seven regressors: Go window with mandibulation (Hit), Go window with no mandibulation (Miss), NoGo window with no mandibulation (CR), NoGo window with mandibulation (FA), reward, post-reward period (initiation reward period, 4 s after rewarding offset, and 5 s after NoGo offset), and mandibulation onset during resting state (the initial, medial, and final resting states) and ITI period (except post-reward period). We used our estimated pigeon HRF to be convolved with the mentioned regressors (see HRF modeling section for more details). In addition, six scan-to-scan estimated motion parameters (Supplementary Fig. 1) were added as nuisance regressors to account for any residual effects of animal movement. Contrasts of interest from first-level analyses were then taken to the higher-level analysis using the mixed-effect model (FLAME1 + 2). To avoid false-positive clustering as mentioned by Eklund et al.[34], higher-level results were thresholded using activation levels determined by $Z > 3.1$ ($p < 0.001$) and FWE cluster significance threshold of $p = 0.05$.

The color discrimination experiment was run in two independent sessions. A two-sample paired $t$ test (implanted in FSL) was used to compare two sessions of the experiment. For each session, an independent GLM was modeled. The results were compared using a two-sample paired $t$ test in FSL. Since there was no significant difference between the two sessions of each run, we modeled both sessions as one GLM model.

An MAD was used to evaluate differences in estimated motion parameters between resting-state and behavioral experiments. MAD is calculated as: $MAD = median(|X_i - \tilde{X}|)$, where $X_i$ is a univariate data set and $\tilde{X} = median(X)$. MAD is a measure of statistical dispersion that is more resilient to outliers in a data set than the standard deviation.

To evaluate the associated stress with the head fixation, statistical testing between days was done with ANOVA for repeated measures with Sidak for adjustment for multiple comparisons (SPSS, version 21). To evaluate the CORT values, we conducted two independent immunoassay tests. A two-tailed paired $t$ test was applied to compare the results of both tests. There was no significant difference between these tests ($p = 0.25$). Repeating the heart rate measurement is not valid since we were able to habituate animals only one time.

**Reporting summary.** Further information on research design is available in the Nature Research Reporting Summary linked to this article.

## Data availability
fMRI data[76] for the color discrimination and the HRF experiments are available at https://data.mendeley.com/datasets/7ps2dvpmjh/1. CORT data are available in the Supplementary information (Supplementary Table 2). A reporting summary for this article is available as a Supplementary information File. Source data are provided with this paper.

## Code availability
FSL software (https://fsl.fmrib.ox.ac.uk/fsl/fslwiki/, version 5.0.9) and MATLAB (2016b, MathWorks, USA) were used to process fMRI and behavioral data, respectively. An instruction to recompile FSL software using pigeon HRF is available in the Supplementary information (Supplementary Methods). Related data processing codes can be found at https://github.com/mehdibehroozi/pigeon_fMRI.

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

## Acknowledgements

This work was supported by the DFG through grant SFB 874 (A1, B5) project number 122679504 and SFB 1280 (A01, A02, A03, A08, and F02) project number 316803389. We thank Abdel Serir for blood sampling. We also thank Bastien Delvaux for his contribution to the ACTH stimulation test experiments.

## Author contributions

O.G. conceived the general idea. M.B., X.H., F.S., and O.G. designed the experiment. T.O. designed the electric equipment and hardware setup. M.B. and M.G. run the experiment. M.B. and X.H. optimized MR sequence. M.B. processed data. X.H. and E.G. were involved in the discussion of data processing. M.B., R.P., R.K., and D.M. designed and conducted the experiments on the stress assessment. M.B., F.S., E.G., X.H., R.P., and O.G. wrote the manuscript. M.B., S.T., and O.G. created the figures. M.T. and N.A. were involved in the discussion.

## Funding

## Competing interests

The authors declare no competing interests.
