## [Peer Review File · Nature Communications]

Reviewer #1 (Remarks to the Author):

I had reviewed this paper already for [REDCATED] and recommended it for publication pending revisions of the figures, which has been followed by the authors.

Reviewer #2 (Remarks to the Author):

The paper by Behroozi and colleagues provides a clear fundamental advancement in the field of avian Neuroscience and more generally cognitive neuroscience by demonstrating a novel paradigm that is able to answer fundamental questions in awake behaving pigeons in the fMRI scanner. The authors designed and report a wonderful setup, optimized fMRI acquisition parameters, and derived a hemodynamic response function for their animal model. In addition, they exemplify the use of this setup to answer a neuroscientific question on the visual system. Importantly the authors also performed experiments that demonstrate that the stress levels of the animals was reduced after habituation. Importantly the authors have addressed a long list of indications from previous reviews and in my view the manuscript can be published in its current form.

Georgios Keliris, PhD
Bio-Imaging Lab
University of Antwerp

Reviewer #3 (Remarks to the Author):

In the manuscript of Behroozi et al. the authors propose a new experimental design that enables them to conduct fMRI measurements during a visual discrimination task in awake, behaving pigeons. After non-human primates and rodents, to my knowledge this study describes the first use of awake, actively-responding birds to map the brain network involved in visual discrimination. I believe that this study constitutes an interesting step forward from the existing methodologies to measure brain activity, and could prove very valuable in deciphering the complex neural networks involved in decision making.

I have a few concerns with the manuscript in its current form however, which I will list below.

1) From the text it is not always entirely clear to me which parts of the described methodology are new innovations and which are adaptations from previous studies. This makes it difficult to judge the scientific contribution that the present study brings.

E.g. line 70 of the Results reads "we established a protocol with which our pigeons were habituated....", while line 371-373 of the M&M reads "The used habituation protocol is a well-established procedure in our lab and has been used in several studies before (Behroozi et al., 2017, 2018b; De Groof et al., 2013)". If the habituation protocol has been established 7 years ago by De Groof et al., 2013, then I don't think this should be brought up as a result from the present study, unless significant changes have been made to the procedure.

Another example of unclarity is in the Abstract line 28/29 where the authors write "...we...optimized the pulse sequence , and reduced repetition times to 2 to 4s" (this is repeated in line 120-123 of the Results). Repetition times of 2s or less appear to be quite common in Spin-Echo fMRI studies including those of the authors (e.g. Poirier et al., 2010 (ref 12); Van Ruijsevelt et al., 2017 (ref 13); Behroozi et al., 2017 (ref 6). In addition I see little difference between the pulse sequence that is used in Behroozi et al., 2017 and in the present study, except for a slightly different Echo Time. If changing the TE from 37 ms to 42 ms is the only 'optimization' of the pulse sequence, I hardly believe that this

warrants mention in the Abstract, or the Results for that matter.

If significant changes have been made to previously established protocols, please present more clearly how the current protocol differs from previous studies and what the significance of the changes are.

2) The authors list a large number of brain regions that are activated during the visual discrimination task. It is expected that many different brain areas are involved in a complex go/no-go procedure, and the manuscript very nicely illustrates the potential of the proposed experimental approach to study higher-order decision making networks. One problem here is that many of the activated brain regions most likely do not contribute to this decision making process, but are involved in secondary tasks (such as motor and somatosensory systems during ingestion of the water). To extract meaningful information on the cognitive function it will be necessary to dissociate the relative contribution of those brain regions involved in decision making from those that do not contribute to this process. It would greatly improve the manuscript if the authors could provide experimental evidence, or at least propose an experimental design where such a dissociation can be made.

Brain activation related to beak movement, ingestion, and reward responses are obviously difficult to filter out without a clever design, but it is unclear to me why the olfactory and auditory system are active in the S+ contrast. The authors shortly discuss the reason for this activation in line 305-308, but is this not activity that you want to prevent? To me this seems a case of bad design that could easily be remedied by presenting similar auditory/olfactory cues during the GO and NO-GO trials. I believe that the proposed methodology would be much more convincing if the authors could demonstrate that they can distinguish activity patterns of interest from unwanted activity patterns.

3) In terms of adapting the presented paradigm to other bird species, I would be interested in the author's take on how well their protocol would work on smaller birds. Pigeons are a valuable model for the visual system, but are relatively big birds, and I am not entirely convinced that their approach can be so "easily adapted to a large number of further bird species" as they mention in line 52. Much research on birds includes work on small passerine birds such as the zebra finch, and it would be very interesting to know if the proposed approach could effectively be used with such small birds. Voxel size for functional imaging in zebra finches is usually in the same range as mice in order to be able to distinguish regions reliably, between 0.15 and 0.25 mm. Considering that smaller voxel sizes suffer more from movement artefacts, do the authors believe that the movement artefacts that their setup allows are small enough for accurate imaging in such small birds?

4) In relation to the previous point, the authors report in Fig 1 E and F the Median absolute deviation (MAD) of translation and rotation. I'm not entirely sure how to interpret these values. Does each circle represent the MAD over a period of image acquisition per individual and do the bar graphs represent the mean of the different MAD's? I think that instead of taking the median of the residuals it would be more informative to report the absolute maximum translation for the different conditions. The authors report this value for the resting condition in line 102-103, but not for the condition where birds are moving their beak a lot, where you would expect the largest translation and rotation. Could you provide the absolute maximum translation and rotation during periods of heavy beak movements? These values for the resting state are reported as 0.048 mm and 0.045 deg. The example bird shown in Figure S2A clearly illustrates short movements in the x and z direction beyond 0.12 mm, so how does that match with this reported value of 0.048 mm? Or is this some averaged value across individuals? Please clarify.

5) The extensive verification of stress levels during the habituation and experimental phases are a very nice addition to the study and appear well thought-out. However, instead of providing the relative

increase of corticosterone (CORT) I would be more interested in the real baseline and experimental CORT concentrations. If baseline levels are already very high, then the relative increase in CORT does not tell us so much about the actual stress level of the animal. In the M&M line 434 a mean baseline CORT concentration of 100.08 ± 79.7 ng/ml is mentioned. I'm no CORT expert, but these baseline levels seem exceptionally high to me. If ACTH induces the maximum stress response this only seems to increase the total CORT concentration by less than 50 ng/ml after 30 minutes, that's only about a 50% increase from baseline levels. In physiological terms that is hardly a stress response. Could the authors verify from other studies what CORT levels are normal for pigeons under relaxed conditions?

I'm also somewhat confused by the numbers in Fig 1C. The percentage change in Fig 1C, is this the percentage increase relative to the baseline of 100 ng/ml? This would mean that on day 1 after 10 minutes the CORT increase of ~ 30 ng/ml is almost 3 times higher than the ACTH induced CORT-concentration after 10 minutes. Or is this 30% of the maximum stress response as indicated in Fig 1B?

Other points:

- Line 47-48: 'avian cognitive revolution' is a bit over the top.
- line 51-53: Sentence is a bit awkward. Rephrase to 'We believe that our approach can easily be...'
- line 70: 'establish': did you establish this in this study, or was this established previously?
- line 142-143: 'Since our birds were quite relaxed during this study and sometimes fell asleep' How did you measure that birds were sleeping inside the bore?
- line 252: 'motion artefacts were reduced to ~ 0.05 mm during mandibulation' Where does this number come from? I did not see any mention of absolute translation during mandibulation in the Results section.
- line 258-259: 'We instead worked with a multiple Spin Echo sequence.' Didn't you use a single-shot SE sequence?
- line 264: 'in plane resolution of $460 \mu\text{m}^2$ ' If you used a spatial resolution of 0.47×0.47 mm then the in plane resolution is $220 \mu\text{m}^2$ not $460 \mu\text{m}^2$.
- line 295: What is meant by 'appetitive'? and what by 'time grain'?
- line 314: 'AVT' I believe the consensus abbreviation for the ventral tegmental area is 'VTA'.
- Throughout the M&M the reference style does not match with the rest of the manuscript (written out instead of numbers).
- Line 384-385: I do not understand this sentence. What is peculiar about the avian skull, and why would head fixation work better in birds than in e.g. rodents? (And what do these peculiarities have to do with how well the head is fixed?).
- line 440: 'comfortably' How do you know it was comfortable for the birds?
- line 461-462: 'intensity of stimuli was 28.9 cd/m^2 '. Is this the luminous intensity value that is referred to as 100% intensity?
- line 540: '3 slices for auditory and visual paradigms' No auditory paradigms have been reported in

the manuscript.

- Fig 1: Please provide individual data points for each individual in Fig 1C and 1D. What is the 'n' for Fig 1D?

- Fig 3: Please consider changing the color shading of the visual and trigeminal pathways (the yellow and red). In combination with the heatmap it becomes difficult to distinguish the shading from the heat map itself.

- Fig 5: Is this connection map based on the activity pattern observed in the current study, and how did the authors obtain the actual connection information? Please provide a reference for this information.

- Fig S2: From which bird and what type of condition is the example translation and rotation information in A? Does this figure represent motion artefacts during activity or inactivity?

- Fig S4: Please align the heat maps in B with the raw images in A. That will make it much easier to compare the two figures.

Response to the reviewers

I am pleased to resubmit the revised version of our Manuscript # NCOMMS-20-17548-T “Event-related functional MRI of awake behaving pigeons at 7T”. We really appreciate the constructive comments and valuable feedback provided by the reviewers in this iteration of the manuscript and in the past and would like to thank them for their efforts. We truly believe that their feedback has improved our manuscript substantially and thus we have taken care to address each of the comments and concerns as detailed as possible. The changes are highlighted in the main text.

We hope that our revisions are satisfactory and look forward to hearing from you soon.

Yours sincerely,

Mehdi Behroozi, on behalf of my co-authors

REVIEWER COMMENTS

Reviewer #1 (Remarks to the Author):

I had reviewed this paper already for [REDACTED] and recommended it for publication pending revisions of the figures, which has been followed by the authors.

We would like to thank the reviewer for this very positive assessment. We are convinced that the points raised by reviewer 1 in the last revision round greatly increased the quality of our manuscript and would thus like to thank her/him for the excellent work.

Reviewer #2 (Remarks to the Author):

The paper by Behroozi and colleagues provides a clear fundamental advancement in the field of avian Neuroscience and more generally cognitive neuroscience by demonstrating a novel paradigm that is able to answer fundamental questions in awake behaving pigeons in the fMRI scanner. The authors designed and report a wonderful setup, optimized fMRI acquisition parameters, and derived a hemodynamic response function for their animal model. In addition, they exemplify the use of this setup to answer a neuroscientific question on the visual system. Importantly the authors also performed experiments that demonstrate that the stress levels of the animals was reduced after habituation. Importantly the authors have addressed a long list of indications from previous reviews and in my view the manuscript can be published in its current form.

We thank reviewer 2 for taking the time to review our manuscript and are happy for this very positive assessment.

Reviewer #3 (Remarks to the Author):

In the manuscript of Behroozi et al. the authors propose a new experimental design that enables them to conduct fMRI measurements during a visual discrimination task in awake, behaving pigeons. After non-human primates and rodents, to my knowledge this study describes the first use of awake, actively-responding birds to map the brain network involved in visual discrimination. I believe that this study constitutes an interesting step forward from the existing methodologies to measure brain activity, and could prove very valuable in deciphering the complex neural networks involved in decision making.

I have a few concerns with the manuscript in its current form however, which I will list below.

1) From the text it is not always entirely clear to me which parts of the described methodology are new innovations and which are adaptations from previous studies. This makes it difficult to judge the scientific contribution that the present study brings.

E.g. line 70 of the Results reads “we established a protocol with which our pigeons were habituated...”, while line 371-373 of the M&M reads “The used habituation protocol is a well-established procedure in our lab and has been used in several studies before (Behroozi et al., 2017, 2018b; De Groof et al., 2013)”. If the habituation protocol has been established 7 years ago by De Groof et al., 2013, then I don’t think this should be brought up as a result from the present study, unless significant changes have been made to the procedure.

Another example of unclarity is in the Abstract line 28/29 where the authors write “...we...optimized the pulse sequence , and reduced repetition times to 2 to 4s” (this is repeated in line 120-123 of the Results). Repetition times of 2s or less appear to be quite common in Spin-Echo fMRI studies including those of the authors (e.g. Poirier et al., 2010 (ref 12); Van Ruijsevelt et al., 2017 (ref 13); Behroozi et al., 2017 (ref 6). In addition I see little difference between the pulse sequence that is used in Behroozi et al., 2017 and in the present study, except for a slightly different Echo Time. If changing the TE from 37 ms to 42 ms is the only ‘optimization’ of the pulse sequence, I hardly believe that this warrants mention in the Abstract, or the Results for that matter.

If significant changes have been made to previously established protocols, please present more clearly how the current protocol differs from previous studies and what the significance of the changes are.

Answer: We are happy to better highlight the innovations of the paper and separate them from adaptations derived from previous studies (see below). In addition, we have improved the abstract to better highlight the major innovations of this paper, as well as complemented the Method section with some references to adapted work in order to make very clear what our contributions were.

Let us first describe our innovation to then talk about the adaptations:

We think it is important to point out that the main methodological innovation of this paper is the creation – to the best of our knowledge for the first time in a bird species - of an active behavioral fMRI platform in awake pigeons, including a demonstration of its performance in an event-based discrimination task. To this end, several innovations were necessary.

First, we chose and established mandibulation as an operant response since it provides two key benefits: *i*) It is the natural behavior of birds during drinking and thus provides an evolutionary ‘prepared’ conditioned response that is easily associated with water as reward. This drastically shortens training of the animals. *ii*) Mandibulation triggers only minor magnetic field distortions that could produce image artefacts during an fMRI experiment. Please see also our response to the reviewer’s question about pigeon skulls and its advantage in terms of reduced expected susceptibility artifacts in behaving birds compared to behaving rodents.

Second, for mandibulation to work as an operant, we created a new MRI compatible piezo-trigger based water reward system and adapted a protocol for pigeon training.

Third, for model-based data processing methods such as the general linear model (GLM), knowledge about the species dependent canonical hemodynamic response function (HRF) is essential to find an accurate activation pattern. Therefore, we experimentally estimated the pigeon HRF based on the experimental BOLD signal, recorded using a single-shot RARE sequence. We used this pigeon HRF to process the color discrimination fMRI time series.

For the next and fourth point we insert this paragraph as a reminder: Another example of unclarity is in the Abstract line 28/29 where the authors write “...we...optimized the pulse sequence , and reduced repetition times to 2 to 4s” (this is repeated in line 120-123 of the Results). Repetition times of 2s or less appear to be quite common in Spin-Echo fMRI studies including those of the authors (e.g. Poirier et al., 2010 (ref 12); Van Ruijsevelt et al., 2017 (ref 13); Behroozi et al., 2017 (ref 6)).

Answer: We agree that it is common to use single-shot Gradient Echo EPI or single-shot (single) Spin Echo EPI in rodent BOLD fMRI with repetition times of 1- 4s. But this is not the case for BOLD fMRI in birds, since the magnetic field inhomogeneities created by air cavities in the avian skull make the use of EPI imaging very challenging.

Recent task fMRI studies by (Poirier et al., 2010) in birds have shown satisfactory BOLD sensitivity and good whole brain coverage with the use of a RARE sequence as alternative to the (single) GE or (single) SE EPI sequences. Usually the RARE sequence in primates and rodents is not used for fMRI but to collect anatomical images as it suffers much less from image distortions introduced by magnetic field inhomogeneities.

In previous studies (Poirier et al., 2010; Van Ruijsevelt et al., 2017), the RARE sequence was used with repetition times of 2 s (which is not the repetition time for volume acquisition) but with a 4x segmented acquisition scheme and not in a single-shot scheme like in our sequence which was introduced in Behroozi et al., 2017. This resulted in a duration of ~6 s to acquire one single 2D image and a duration of 8 s (4x2 s) for the acquisition of one volume. The repetition time between two volumes was therefore 8s, too.

Following our work in (Behroozi et al., 2017), we now have introduced a single-shot RARE with 43 spin echoes, an inter-echo time of 4 ms and a repetition time of 2 - 4 s. Therefore, the duration to acquire one 2D image was only $43 * 4 \text{ ms} = 172 \text{ ms}$ in our experiments and the repetition time between two volumes was 2 - 4 s. The reduced image acquisition time (a few tens of milliseconds against many seconds in past bird RARE fMRI literature) makes our RARE fMRI sequence much more robust against animal motion or magnetic field variations induced by breathing or active mandibulations. In addition, this short 2D image acquisition time combined with shorter volume repetition times (2 s and 4 s) is better suited for recording brain BOLD response during event-

based fMRI protocols. In addition, we remark that the biggest innovation/novelty with the single-shot RARE used in this study lies in the application of the single-shot RARE adapted from Behroozi et al., 2017 (resting-state) to task-based fMRI in pigeons for the first time.

To conclude, the task-based fMRI protocol proposed in this study is definitely a fourth significant technological progress in the context of bird fMRI. We have made this more prominent by slightly modifying the abstract (see above).

To highlight these four points, we slightly modified the abstract (highlighted with bold font), while removing unnecessary details as noted by the reviewer (see further below).

The abstract now reads:

*“Animal-fMRI is a powerful method to understand neural mechanisms of cognition, but it remains a major challenge to scan actively participating small animals under low-stress conditions. Here, we present an event-related functional MRI platform in awake pigeons **using single-shot RARE fMRI** to investigate the neural fundamentals for visually-guided decision making. We established a **head-fixated Go/NoGo paradigm** which the animals quickly learned under low-stress conditions. **The animals were motivated by water reward and behavior was assessed by logging mandibulations during the fMRI experiment with close to zero motion artifacts over hundreds of repeats.** To achieve optimal results, we characterized the species-specific hemodynamic response function. As a proof-of-principle, we run a color discrimination task and discovered differential neural networks for Go-, NoGo-, and response execution-phases. Our findings open the door to visualize the neural fundamentals of perceptual and cognitive functions in birds - a vertebrate class of which some clades are cognitively on par with primates.”*

The Method section now reads:

*“These slices were acquired using a single shot multi-slice RARE sequence **adapted from Behroozi et al.,⁶** with following parameters: $TR = 4000$ ms ($TR = 2000$ ms for visual paradigms which were used to characterize pigeon HRF), $TE_{eff} = 41.58$ ms (**the TE_{eff} was chosen such as to match $T2$ of pigeon brain²¹**),”*

We now list our incremental adaptations:

First, we adapted the established habituation protocol that has been used for short functional scans (usually less than 20 min) in several studies (Behroozi et al., 2017; De Groof et al., 2013). However, the existing protocol needed to be validated for prolonged periods of fMRI scanning, since task based BOLD fMRI studies often require extended scanning sessions of around one-hour duration. Therefore, we carefully monitored the animal’s behavior, resulting moving parameters and stress related parameters (i.e. CORT levels and heart rates, see below). Our data indicate that the habituation protocol is suited to accustom pigeons to longer fMRI experiments, which is in our opinion very relevant to report.

We slightly altered the main text to make clear that the general protocol used was indeed like in previous studies but that it was validated for much longer fixation periods.

The Results section reads: “

*When fully conscious animals are scanned, stress levels and, concomitantly, motion artifacts can rise. We therefore **adapted** a protocol with which our pigeons were habituated to a custom-made restrainer (Fig. 1A, S1, and Movie S1) **for more than an hour.**”*

The Method section reads:

“The used habituation protocol is a well-established protocol in our laboratory^{6,7,21} for short fixation periods but validated in this study for longer periods of animal fixation above an hour.”

For the next point we insert this paragraph as a reminder: In addition I see little difference between the pulse sequence that is used in Behroozi et al., 2017 and in the present study, except for a slightly different Echo Time. If changing the TE from 37 ms to 42 ms is the only ‘optimization’ of the pulse sequence, I hardly believe that this warrants mention in the Abstract, or the Results for that matter.

Answer: We generally agree with the reviewer that the differences between the cited RARE sequence and the used RARE parameters are not big and we thank her/him for bringing up this issue. We have concluded that the details on repetition times in the abstract should be removed (see above). Combined with some more text reorganization (see below), we think that we have reached a more structured and concise report of our study.

Nevertheless, we want to point out that in addition to adapting the number of slices, we have made few but very significant modifications from our RARE sequence presented in Behroozi et al., 2017: the repetition time (TR) was increased from 2s to 4s to reduce animal stress, the effective echo time (TE_{eff}) was increased to 41.58 ms to optimize BOLD sensitivity and we had to do a saturation of the eye signals to remove some image artifacts in brain regions close to eyes. We are convinced that these modifications have vastly improved the quality of our fMRI results. All those improvements are clearly described in the discussion and indicated in the methods part. Behroozi et al., 2017 is now cited in the method part as well.

Seen from this perspective, and to conclude our response, the task-based fMRI protocol presented in this study is new, performed better, and is more robust than the segmented variant published so far and we show that its usage can be extended from block-designs to event-based designs. Finally, we could show that the single-shot RARE does perform very well even while our pigeons are behaving. We now also discuss these aspects in the discussion (see below for changes).

We slightly altered the main text to make the adaptations clear. The discussion now reads:

*“To achieve whole brain coverage during scanning, we had to refrain from using a conventional gradient echo sequence at 7T used for birds in the past, since it restricts the field of view due to strong local magnetic field inhomogeneities caused by air cavities in the bird skull^{5,12}. We instead worked with a **RARE** sequence. A 4X segmented RARE sequence has been used earlier to map responses to auditory¹² and visual stimuli in birds⁷ with repetition time of 8 s and an echo time of 60 ms. However, this only allows running block design experiments and does not provide sufficient sensitivity and temporal resolution for event-related designs. **In the current study, we thus***

used the single-shot RARE sequence adopted from Behroozi et al⁶ for our experiments, which overcomes these limitations and permits recording whole brain responses with an in-plane resolution of 0.47x0.47 mm² within less than 4 s during an active discrimination task. Although the repetition time could have been kept at 2 s for the 11 slices covering the whole brain as in Behroozi et al.⁶, we refrained from doing so to further reduce animals' stress levels by reducing gradient noise and heat caused by radio frequency pulses. In addition, we optimized BOLD sensitivity of this RARE sequence by adjusting the effective echo time to 42 ms, a value close to the typical T2 relaxation times of pigeon brain tissue²¹. Since pigeons possess relatively large eyes and moving them can induce artifacts in the brain regions close to them, we used saturation slices to saturate eye signals which helped to remove some image artifacts. However, we note that, compared to BOLD fMRI EPI-based sequences, the RARE sequence is much less sensitive to magnetic field fluctuations created by breathing or mandibulations and does suffer only little from geometrical distortions. These changes enabled our event-related color discrimination study in pigeons."

2) The authors list a large number of brain regions that are activated during the visual discrimination task. It is expected that many different brain areas are involved in a complex go/no-go procedure, and the manuscript very nicely illustrates the potential of the proposed experimental approach to study higher-order decision making networks. One problem here is that many of the activated brain regions most likely do not contribute to this decision making process, but are involved in secondary tasks (such as motor and somatosensory systems during ingestion of the water). To extract meaningful information on the cognitive function it will be necessary to dissociate the relative contribution of those brain regions involved in decision making from those that do not contribute to this process. It would greatly improve the manuscript if the authors could provide experimental evidence, or at least propose an experimental design where such a dissociation can be made.

Answer: We thank the reviewer for giving us the possibility to clarify the potential of our setup to run complex experiments on cognitive abilities. We thus updated our discussion and added a paragraph to the supplementary materials to propose exemplary changes in the experimental setup, allowing to disentangle more precisely different components of the decision-making process.

The paragraph in the supplementary materials reads:

"Additional procedures to disentangle the neural activity of different components during a Go/NoGo task"

We demonstrated that our experimental setup is well suited to reveal neural activation patterns during a visually guided Go/NoGo task. Future studies could differentiate further neurocognitive components by some modifications of the current protocol. As an example: Processing of the Go stimulus consists of following steps: encoding of the stimulus; reward-based stimulus-response association, decision making, and motor output. In addition, associative learning studies demonstrated that reward expectation can severely influence the representation of sensory stimuli^{1,2}. For example, if a Go stimulus reward has been coupled with a specific smell and/or sound, a pure visual presentation of the Go stimulus alone can elicit activation in the auditory, olfactory and somatosensory systems even if no auditory/olfactory stimuli are present and the reward ingestion

has not begun yet.

On the other hand, processing of NoGo stimuli mostly requires the relevant sensory system for stimulus encoding and inhibitory systems to prevent the animals' response.

The method of choice to differentiate between different components in an fMRI data set is subtraction method. Using the Go vs. NoGo contrast (as we did in the current study) is well suited to identify active brain regions during the Go stimulus phase, which, as we have shown, are located in the motor execution, decision making, reward expectation and possibly further sensory areas such as somatosensory (due to ingestion), auditory (because of extra sound) or olfactory (reward smell) systems, which were associated with the reward. Disentangling different components of these systems is much more challenging and requires changing the Go/NoGo paradigm in a way that either enables modeling of each component separately or balances events in both Go and NoGo trials. In the current study, we found that animals were mandibulating randomly during the ITI. By subtracting the activation pattern recorded during random mandibulations from the Go activation pattern, we were able to eliminate the motor component (beak movement) from our results. In addition, we designed our experiment in a way that the reward was only delivered after Go stimulus offset to disassociate the ingestion effect.

To eliminate reward associated components like reward smell or sound during reward presentation as well as reward expectation, we would suggest a symmetrically reward Go/NoGo experiment³. This experiment is more or less identical to the paradigm used in our study with the exception that reward associated sensory stimuli as well as factors like reward expectation are present in both Go and NoGo trials and can thus be subtracted from the data. The remaining activity pattern should thus solely represent the decision-making component.”

The discussion now reads:

*“In our task, the delivery of the water reward was also associated with multiple non-visual cues like the soft sound of the water pipe valve, the smell of the inflowing water as well as the trigeminal and tactile feedback during water consumption. **It has been shown that such activities in early sensory cortices of primates reflects perceptual experience, rather than presence of an external sensory stimulation**^{41,42}. A study by Meyer et al.⁴¹ demonstrated that visual stimuli, which are associated with auditory stimuli, can activate the auditory system in human brain even in absence of the auditory stimulus. In addition, Zhou and Fuster⁴² could show that during a visual-haptic associative learning task in monkey, somatosensory neurons would also respond to visual stimuli, when the visual stimulus has been associated with a haptic stimulus before. In line with **these results**, we found activity in multiple primary and associative auditory (field L1-3, NCM, MC)^{38,43} and olfactory (BO, CPP, MOT, TuO)^{44,45} areas, likely caused by an association of auditory and olfactory stimuli to the reward. We also identified BOLD responses in the trigeminal and tactile pallial areas (HA, NFT)⁴⁶ that possibly result from somatosensory feedback during reward ingestion. In addition, the polysensory MIVm that receives input from various sensory and limbic structures was active³⁸. Thus, the post-choice presence of multiple reward-associated cues created multisensory activations at forebrain level. **To prevent such activations, future studies could make use of a symmetrically rewarded Go/NoGo design**⁴⁷ which balances reward expectation and association components which could thus be removed from the analysis (see supplementary for more details).”*

Brain activation related to beak movement, ingestion, and reward responses are obviously difficult to filter out without a clever design, but it is unclear to me why the olfactory and auditory system are active in the S+ contrast. The authors shortly discuss the reason for this activation in line 305-308, but is this not activity that you want to prevent? To me this seems a case of bad design that could easily be remedied by presenting similar auditory/olfactory cues during the GO and NO-GO trials. I believe that the proposed methodology would be much more convincing if the authors could demonstrate that they can distinguish activity patterns of interest from unwanted activity patterns.

Answer: Please see the new paragraphs above for the supplementary materials and the discussion.

3) In terms of adapting the presented paradigm to other bird species, I would be interested in the author's take on how well their protocol would work on smaller birds. Pigeons are a valuable model for the visual system, but are relatively big birds, and I am not entirely convinced that their approach can be so "easily adapted to a large number of further bird species" as they mention in line 52. Much research on birds includes work on small passerine birds such as the zebra finch, and it would be very interesting to know if the proposed approach could effectively be used with such small birds. Voxel size for functional imaging in zebra finches is usually in the same range as mice in order to be able to distinguish regions reliably, between 0.15 and 0.25 mm. Considering that smaller voxel sizes suffer more from movement artefacts, do the authors believe that the movement artefacts that their setup allows are small enough for accurate imaging in such small birds?

Answer: We are sure that the transfer to another species and most importantly songbirds like zebra finches (as a major model species) is possible. The reviewer is right that in small species motion artifacts are a major concern, but this concern can be overcome by few changes to our protocol.

We have used an in-plane field of view (FOV) of 0.47x0.47 mm² with 64x64 pixels per image. In zebra finches, a reduced in-plane FOV of only 16x16 mm² can be used (Van Ruijssevelt et al., 2018). By simply changing the FOV of our single-shot RARE sequence to adapt the FOV of zebra finches would result in a BOLD fMRI sequence with an in-plane resolution of \$(0.47/64) \times (0.47/64) = 0.25 \times 0.25\$ mm². This in-plane image resolution would already be sufficient to run a broad range of BOLD fMRI studies in the zebra finch brain. For example, Van Ruijssevelt et al. (2018) reported successful fMRI scans of the auditory system with a coarser in-plane resolution of 0.25x0.5 mm². Reducing the slice thickness to 0.75 mm as used in (Van Ruijssevelt et al. 2018) is definitively possible with our MRI scanner despite the fact that our scanner is equipped with hardware optimized for bigger animals like rats (and pigeons). Achieving a slice thickness of 0.6 mm would still be possible at the price of a slightly increased effective echo time. A single-shot RARE protocol with significantly lower slice thickness and in-plane resolution at constant effective echo times will be even easier to reach if one were to use commercially available MRI scanner hardware dedicated to the size of mice (i.e. zebra finches) imaging: a smaller resonator for shorter RF pulses and a stronger imaging gradient system with reduced switching gradient delays for faster imaging. To conclude, the single-shot RARE sequence with parameters scaled for satisfactory image resolution in small bird species will allow BOLD fMRI in the zebra finch brain with good robustness towards motion artefacts and magnetic field inhomogeneities.

As a second argument, we report a maximum of 0.05 mm head displacement for our pigeons. This is 6 times less motion displacement compared to a reported awake zebra finch fMRI setup (Van

Ruijssevelt et al. 2017). Certainly, the much better head fixation system of our pigeons is to be attributed to the use of a pedestal which is then tightly screwed to the animal bed and firmly constrains head motion. In the zebra finch setup, no head-socket was used. Therefore, it is reasonable to consider a maximum motion displacement of 0.05 mm for zebra finches using our platform (when adapted to zebra finches). With an adapted in-plane fMRI image resolution of $0.25 \times 0.25 \text{ mm}^2$, the expected motion displacement would represent 1/5th of the shortest voxel dimension. This should allow high quality fMRI results in the zebra finch.

Finally, if we look at the rodent field, we note that fMRI in awake mice at in-plane image resolutions as high as $0.1 \times 0.1 \text{ mm}^2$ have been successfully reported (Desai et al., 2011). Head motion in awake head fixed mice can therefore obviously be sufficiently controlled to allow fMRI. As noted by the reviewer, the zebra finch brain has a size comparable to the mouse brain and there are no obvious arguments why motion would be more difficult to control in the zebra finch. In fact, we even think that the anatomy of the avian skull could make it easier to fixate birds compared to rodents, so that motion should be less a problem in zebra finches than in mice (see comments below).

To better illustrate the transferability of our pigeon platform to smaller bird species we have added following sentence in the discussion:

"In particular, the proposed platform should scale well to smaller bird species (like the zebra finch). Assuming conservatively a maximum motion displacement of 0.05 mm and directly adapting the described single-shot RARE protocol to the FOV of the zebra finch brain ($16 \times 16 \text{ mm}^2$)¹³ would result in a imaging in-plane resolution of $0.25 \times 0.25 \text{ mm}^2$ with motion displacements smaller than a fifth of the voxel size. These numbers should allow a broad range of fMRI investigations."

4) In relation to the previous point, the authors report in Fig 1 E and F the Median absolute deviation (MAD) of translation and rotation. I'm not entirely sure how to interpret these values. Does each circle represent the MAD over a period of image acquisition per individual and do the bar graphs represent the mean of the different MAD's?

Answer: The reason for reporting MAD values was due to a request of reviewer 2 in the first round of the revision process. Since we have three translation and three rotation parameters, we used MAD to get a single value for rotation and one for translation of whole brain movement during each experiment. We concatenated all three translation or rotation parameters over each period of image acquisition of each animal to get a single vector for both parameters - translation and rotation- which we used to calculate MAD. The output was the absolute deviation of translation or rotation from the median for each experiment. In Figure 1E and F, each circle represents a single MAD value per individual during different experiments during image acquisition. The bar graphs represent the mean of MAD values of rotation and translation of all animals during different experiments.

I think that instead of taking the median of the residuals it would be more informative to report the absolute maximum translation for the different conditions. The authors report this value for the resting condition in line 102-103, but not for the condition where birds are moving their beak a lot,

where you would expect the largest translation and rotation. Could you provide the absolute maximum translation and rotation during periods of heavy beak movements?

Answer: We apologize for the missing information. We now report maximum absolute parameters for the color discrimination experiment (where the strongest beak movements occurred) and the resting state experiment in Fig. S2.

In addition, we figured that there was a mistake in lines 102-103. The reported values were for the color discrimination experiment, not for the resting state. We thus updated the text by adding motion parameters for both color discrimination and resting state experiments.

The text now reads:

“The maximum absolute translation and rotation during the color discrimination experiment were 0.048 mm and 0.045°, and 0.041 mm and 0.043° during the resting-state fMRI time series, respectively (Fig. 1, E, F and Fig. S2).”

These values for the resting state are reported as 0.048 mm and 0.045 deg. The example bird shown in Figure S2A clearly illustrates short movements in the x and z direction beyond 0.12 mm, so how does that match with this reported value of 0.048 mm? Or is this some averaged value across individuals? Please clarify.

Answer: We agree with the reviewer that these values were not sufficiently explained/sub optimally chosen and we apologize for this. The data in figure S2A showed the estimated motion parameters during the color discrimination task of one bird. Since we used an established human fMRI software (FSL) to process the data, we had to upscale the voxel size in our pigeon data set to the voxel size level used in humans before applying any processing steps. The reported values in Figure S2 were calculated after upscaling voxel size by a factor 10 and are thus 10 times bigger than the actual movement. Although we mentioned this in the figure captions, we now downscaled the values in figure S2A again to keep them constant between main text and supplementary.

Find below the updated figure and figure captions.

Figure S2- Estimated motion parameters of an individual pigeon during active color discrimination. (A) Top row shows the translational (in mm), bottom row rotational movement parameters (in deg.) of a single individual. Parameters were estimated by a 3D rigid body model with six degrees of freedom for translation (x, y, and z direction) and rotation (pitch, roll, and yaw). (B) Mean of maximum absolute rotation and translation parameters over all pigeons (n=8) during resting-state and color discrimination experiments. Circles represent single individuals.

5) The extensive verification of stress levels during the habituation and experimental phases are a very nice addition to the study and appear well thought-out.

However, instead of providing the relative increase of corticosterone (CORT) I would be more interested in the real baseline and experimental CORT concentrations.

Answer: We now provide these values in separate table as supplementary material.

Table S1 – Average plasma corticosterone values (ng/ml) \pm SEM of four individual pigeons on different habituation days.

	Day1 0 Minutes	Day1 10 Minutes	Day4 0 Minutes	Day4 10 Minutes	Day7 0 Minutes	Day7 10 Minutes
P _{Cort} 1	247.8 \pm 11.3	258.8 \pm 12.8	213.9 \pm 9.7	196.8 \pm 7.7	250.5 \pm 14.0	241.9 \pm 13.8
Change Δ		11.1		-17.1		-8.5
P _{Cort} 2	110.6 \pm 3.4	117.8 \pm 6.0	120.1 \pm 4.1	115.0 \pm 5.1	123.3 \pm 5.9	126.2 \pm 4.7
Change Δ		7.3		-5.1		2.9
P _{Cort} 3	56.2 \pm 3.7	85.3 \pm 5.1	72.0 \pm 1.8	76.0 \pm 5.3	67.2 \pm 7.0	68.9 \pm 6.2
Change Δ		29.2		4.0		1.7
P _{Cort} 4	47.2 \pm 3.0	68.8 \pm 3.4	59.2 \pm 4.6	67.3 \pm 3.2	72.3 \pm 6.2	76.4 \pm 7.0
Change Δ		21.6		8.2		4.1
Average Δ		17.3 \pm 4.99		-2.5 \pm 5.6		0.1 \pm 2.9

If baseline levels are already very high, then the relative increase in CORT does not tell us so much about the actual stress level of the animal.

Answer: The reviewer is correct in pointing that out. We now have changed relative values to the individual increase/decrease of the measured CORT values to increase the comparability between the ACTH stimulation test results over time and the experimental data (Figure 1 C, D in main text).

Fig. 1- Platform for awake pigeon fMRI. (A) Head restrained pigeons were positioned in the scanner bore center and stimuli were presented using fiber optics connected to a light stimulator outside the scanner room. Animal responses were registered using a piezo-electric sensor mounted under the lower jaw. (B) Corticosterone (CORT) increase in response to the adrenocorticotrophic hormone (ACTH) stimulation test (n = 3; Mean \pm SEM). Mean CORT concentration increases over time after ACTH administration. (C) Absolute change in plasma corticosterone on habituation days D1, D4, and D7. Absolute change in CORT was significantly lower on habituation day 4 and

day 7 compared to habituation day 1. Bars = mean values of absolute corticosterone level change ($n = 4$). **(D)** Heart rate during the habituation procedure on day 1, 4, and 7 ($n = 4$). Heart rate was significantly lower on habituation day 7 compared to habituation day 1 (dotted line = baseline at rest). **(E)**, **(F)** Median absolute deviation (MAD) of estimated motion parameters (translation and rotation) over resting state and active decision task ($n = 8$). **(G)** Example of activation clusters caused by 2 s visual stimulation (calculated for an exemplary pigeon brain for the first run) to estimate the hemodynamic response function parameters. The activation mask from individual pigeons was applied to the other runs to extract the time course of hemodynamic responses within each ROI. **(H)** Left curve shows the BOLD signals (HDR) for different visual stimuli (average of 5 subjects) while the right curve shows the best fitted pigeon hemodynamic response function ($\alpha_1 = 7.71$, $\alpha_2 = 11.48$, $\beta_1 = 1.74$, $\beta_2 = 0.74$, $c = 0.25$). **(I)** Estimated parameters of HDR for different colors. Hemodynamic responses did not differ between colors. Each circle in C, D, E, F, and I represent a single value per individual during different experiments. Abbreviations: ACTH: adrenocorticotrophic hormone; CORT: corticosterone; BOLD: Blood-oxygen-level-dependent; H: height; T: time-to-peak; W: full-width at half maximum; A: anterior; P: posterior; L: left; R: right.

In the M&M line 434 a mean baseline CORT concentration of 100.08 ± 79.7 ng/ml is mentioned. I'm no CORT expert, but these baseline levels seem exceptionally high to me.

Answer: Indeed, different factors have an impact on baseline values of plasma CORT such as the sex and reproduction state of the animal (Chatelain et al. 2018), plumage coloration (Angelier et al. 2018), diurnal variation (Joseph & Meier 1973; Rintamäki et al. 1986; Westerhof et al. 1994), and seasonal variation (Rintamäki et al. 1986). Further, housing of the birds also has an impact on baseline CORT levels (Rintamäki et al. 1986; Pakkala et al. 2013). Besides considerable inter-individual differences, the ability to cope with these influences also varies on an individual level, resulting in animals that return to baseline levels while other individuals show persistent elevated CORT levels. However, it has been shown that a comparable CORT increase can be elicited when a stressor is applied for the second time, even if baseline CORT levels are elevated (Angelier et al. 2016). Thus, the elevated and distributed baseline levels we report in our study might reflect these different influences. Alongside the above-mentioned physiological aspects, the methodology used to quantify the CORT response impact the scale of individual results leading to some variations of reported concentrations in the literature Westerhof (1994). Despite these shortcomings, given the overall reduction of CORT levels in conjunction with the reduced heart beat rates over time, we are confident that the reported values do support the claim that the stress-related CORT responses are reduced due to the applied habituation protocol.

If ACTH induces the maximum stress response this only seems to increase the total CORT concentration by less than 50 ng/ml after 30 minutes, that's only about a 50% increase from baseline levels. In physiological terms that is hardly a stress response. Could the authors verify from other studies what CORT levels are normal for pigeons under relaxed conditions?

Answer: As outlined above, it is difficult to refer to the "normal pigeon under relaxed conditions" since values in the literature are astonishingly variable. If required, we are happy to add a larger overview of the relevant literature, the previously reported values and used methods in the supplementary. It is, obviously, conceivable that higher dosages of injected ACTH might lead to a more

pronounced CORT response. Our chosen dosage to reliably detect changes in the CORT levels followed the suggestions made by Lumeij et al. (1987).

I'm also somewhat confused by the numbers in Fig 1C. The percentage change in Fig 1C, is this the percentage increase relative to the baseline of 100 ng/ml? This would mean that on day 1 after 10 minutes the CORT increase of ~ 30 ng/ml is almost 3 times higher than the ACTH induced CORT-concentration after 10 minutes. Or is this 30% of the maximum stress response as indicated in Fig 1B?

Answer: We thank the reviewer for pointing out the misleading presentation of our data and apologize for the resulting confusion. The data reported were the relative changes of the CORT concentration for each individual. However, according to the suggestion made above, we now report the increase/decrease of the measured CORT values instead of the percentage of individual change in CORT (Figure 1 in main text). Further, we now indicate the individual data points in the figure to enable a direct comparison between the changes induced by the ACTH stimulation test and the measured values during the habituation protocol.

Fig 1: Please provide individual data points for each individual in Fig 1C and 1D. What is the 'n' for Fig 1D?

Answer: Individual data points have now been included in both panels of figure 1. We used 4 animals for the evaluation of stress and added this information to the figure captions (see above).

Other points:

- Line 47-48: 'avian cognitive revolution' is a bit over the top.

Answer: We now toned it down and write:

"In addition, the recent findings on avian cognition demonstrated that especially corvids and parrots are cognitively on par with primates¹⁹."

- line 51-53: Sentence is a bit awkward. Rephrase to 'We believe that our approach can easily be...'

Answer: We rephrased the sentence. It now reads:

"We believe that our approach can easily be adapted to a large number of further bird species and possibly beyond"

- line 70: 'establish': did you establish this in this study, or was this established previously?

Answer: We apologize for not being clear about this point and changed the text accordingly (compare our response to comment 1).

- line 142-143: 'Since our birds were quite relaxed during this study and sometimes fell asleep' How did you measure that birds were sleeping inside the bore?

Answer: We used an MRI compatible camera to monitor the animals during the experiments. During phases, where the animals were not required to respond, they often closed their eyes, showed a reduced muscle tension (e.g. relaxed neck muscles indicated by puffed up feathers) and showed no random movements (awake animals occasionally move their feet or mandibulate from time to time). However, the reviewer is right that we do not have a scientific proof that the animals were really sleeping, and we thus updated the text. The sentence now reads:

“Since our birds closed their eyes and sometimes appeared to have fallen asleep, we had to increase their attention to the task (Fig. S5).”

- line 252: ‘motion artefacts were reduced to ~0.05 mm during mandibulation’ Where does this number come from? I did not see any mention of absolute translation during mandibulation in the Results section.

Answer: We apologize for the missing information. We updated Figure S2 with the requested information about maximum absolute motion parameters during both color discrimination and resting state experiments. For more details, please refer to our answer to comment 4.

- line 258-259: ‘We instead worked with a multiple Spin Echo sequence.’ Didn’t you use a single-shot SE sequence?

Answer: We have used a single-shot RARE sequence and we admit that our use of the expression “multiple Spin Echo sequence” might have been confusing since the expression “multiple Spin Echo sequence” might be better known in the context of acquiring fast multiple SE images with different T2 weightings. We have now consistently replaced the expression “multiple Spin Echo sequence” with the expression “RARE sequence” to avoid any ambiguity.

We had used the expression “multiple Spin Echo sequence” as a somewhat loose synonym to refer to the RARE sequence since a RARE sequence is effectively made of a succession of spin echoes (43 spin echoes in our single-shot RARE sequence), each spin echo being differently phase encoded. In single-shot mode, all the phase encodings required to reconstruct one 2D RARE image are acquired following one unique excitation pulse.

Please refer to our answer to comment 1 for more details.

- line 264: ‘in plane resolution of 460 μm^2 ’ If you used a spatial resolution of 0.47 x 0.47 mm then the in plane resolution is 220 μm^2 not 460 μm^2 .

Answer: That was indeed misleading, and we apologize for this mistake. We have updated the text as follows:

“In the current study, we thus used the single-shot RARE sequence adopted from Behroozi et al.⁶ for our experiments, which overcomes these limitations and permits recording whole brain responses with an in-plane resolution of 0.47x0.47 mm^2 within less than 4 s during an active discrimination task”

- line 295: What is meant by ‘appetitive’? and what by ‘time grain’?

Answer: In the behavioral sciences, appetitive means positive consummatory reward based. In our

study, the S+ is associated with such a positive reward (water). In contrast, some studies associate their S+ with an aversive reward (or punishment: electric shock, air puff), which serves the same purpose but results in avoidance of this cue. Since we believe that this term is well known in the field, we would prefer to leave it as is.

Time grain means time resolution, but we have changed the sentence to make it more accessible:

“We visualized the avian neural network for an appetitive Go/NoGo color discrimination task with a time resolution of 4 s.”

- line 314: ‘AVT’ I believe the consensus abbreviation for the ventral tegmental area is ‘VTA’.
Answer: We now switched to the anglicized term VTA.

- Throughout the M&M the reference style does not match with the rest of the manuscript (written out instead of numbers).

Answer: We would like to thank the reviewer for pointing out this mistake it. We have corrected all references and they should now all be in line with the Nature Communication reference style.

- Line 384-385: I do not understand this sentence. What is peculiar about the avian skull, and why would head fixation work better in birds than in e.g. rodents? (And what do these peculiarities have to do with how well the head is fixed?).

Answer: There are two main reasons why a good fixation in birds is more effective in reducing motion artifacts during an fMRI experiment in comparison to rodents. First, during evolution, birds developed a skull structure, which is notably different from mammals (Dumont 2010). Likely as an adaptation to flight, bone density increased, maximizing bone strength and stiffness. At the same time, birds included pneumatized spaces (air cavities) into their skull structure to reduce absolute bone mass and volume and thus weight (Currey 2003). This bone structure is perfectly suited to give hold to the screws carrying the implant with which the bird is fixated in the scanner. Second, rodent masticatory muscles (jaw muscles) cover the whole head and run around the skull (Cox and Jeffery 2011). Any movement of these muscles (e.g. during mandibulations) can change the magnetic field homogeneity, which could cause inhomogeneity distortions. These distortions can be detected as motion artifacts. In contrast, the muscles controlling lower beak movements in birds do not run over the skull and are only attached to few connection points on the skull directly below the lower jaw - birds can and do not masticate (Jones et al., 2019). Both, “screw friendly” bone structure and the lack of mastication induced motion artifacts are clear advantages of the avian skull when performing fMRI experiments.

Figure 1- Lateral view of 3D reconstruction of jaw closing muscles of (a) pigeon and (b) rat. Pictures are from Cox and Jeffery 2011 and Jones et al., 2019.

The sentence now reads:

“It has been shown to be especially effective in bird species due to their specific bone structure⁶³ and lack of mastication muscles⁶⁴.”

- line 440: ‘comfortably’ How do you know it was comfortable for the birds?
 Answer: The reviewer is obviously right that we do not know if the animals were really feeling comfortable since we cannot read their minds and rely on stress measures and their behavior that was constantly monitored by means of a camera. In addition, there was no obstacle under the beak, such that the pigeons were able to freely mandibulate whenever they wanted to. However, we rephrased the sentence to make clear that we do not have scientific proof to confirm this. The sentence now reads:

“Animals’ behavior was monitored carefully during all training phases by observation or camera to ensure that they could open their lower jaw (mandibulation) as the operant response without any obstruction.”

- line 461-462: ‘intensity of stimuli was 28.9 cd/m²’. Is this the luminous intensity value that is referred to as 100% intensity?

Answer: Yes, 100% light intensity was 28.9 cd/m². We now have added this information to the text.

“In addition, to ensure that pigeons were attending to the light color but not to intensity, two different light intensities for both colors (20% or 100% of the maximum light intensity 28.9 cd/m²) were used during the experiment.”

- line 540: ‘3 slices for auditory and visual paradigms’ No auditory paradigms have been reported in the manuscript.

Answer: We deleted “auditory”.

- Fig 1: Please provide individual data points for each individual in Fig 1C and 1D. What is the ‘n’ for Fig 1D?

Answer: We apologize for the missing information. We used four pigeons to evaluate CORT levels

during habituation. We changed the respective figures and added the missing information.

- Fig 3: Please consider changing the color shading of the visual and trigeminal pathways (the yellow and red). In combination with the heatmap it becomes difficult to distinguish the shading from the heat map itself.

Answer: We have changed the colors in figure 3 according to the reviewer's suggestion.

- Fig 5: Is this connection map based on the activity pattern observed in the current study, and how did the authors obtain the actual connection information? Please provide a reference for this information.

Answer: Telencephalic connectivity's were listed and analyzed in the pigeon connectome study of Shanahan et al. (2013).

- Fig S2: From which bird and what type of condition is the example translation and rotation information in A? Does this figure represent motion artefacts during activity or inactivity?

Answer: We apologize for the missing information. The data shows the movement parameters from a single animal during the color discrimination experiment. We now have updated the figure

captions.

- Fig S4: Please align the heat maps in B with the raw images in A. That will make it much easier to compare the two figures.

Answer: Done

References:

Angelier, F., Parenteau, C., Trouvé, C., and Angelier, N. Does the stress response predict the ability of wild birds to adjust to short-term captivity? A study of the rock pigeon (*Columbia livia*). *Royal Society Open Science* 3: 160840 (2016).

Angelier, F., Parenteau, C., Trouvé, C., and Angelier, N. The behavioural and physiological stress responses are linked to plumage coloration in the rock pigeon (*Columbia livia*). *Physiology & behavior* 184, S. 261–267 (2018).

Chatelain, M., Gasparini, J., Frantz, A., and Angelier, F. Reproduction impairments in metal-polluted environments and parental hormones: No evidence for a causal association in an experimental study in breeding feral pigeons exposed to lead and zinc. *Ecotoxicology and Environmental Safety* 161, S. 746–754 (2018).

Cox, P.G., Jeffery, N. Reviewing the Morphology of the Jaw-Closing Musculature in Squirrels, Rats, and Guinea Pigs With Contrast-Enhanced microCT. *Anat Rec*, 294(6):915-28 (2011).

Currey, J.D. The many adaptations of bone. *Journal of Biomechanics* 36:1487–95 (2003).

Desai, M.I., Kahn, U., Knoblich, J., Bernstein, H., Atallah, A., Yang, N., Kopell, et al. “Mapping Brain Networks in Awake Mice Using Combined Optical Neural Control and FMRI.” *Journal of Neurophysiology* 105,3: 1393–1405 (2011).

Dumont, ER. Bone density and the lightweight skeletons of birds. *Proceedings of the Royal Society B Biological Sciences* 277:2193–8 (2010).

Lumeij, J. T., Boschma, Y., Mol, J., Kloet, E. R. de, and Van den Brom, W. E. Action of acth upon plasma corticosterone concentrations in racing pigeons (*Columba livia domestica*). In: *Avian Pathology* 16, S. 199–204 (1987).

Jones M.E.H., Button D.J., Barrett P.M., Porro L.B. Digital dissection of the head of the rock dove (*Columba livia*) using contrast-enhanced computed tomography. *Zoological Letters* 5, 17 (2019).

Joseph, M.M. and Meier, A.H. Daily rhythms of plasma corticosterone in the common pigeon, *Columba livia*. In: *General and Comparative Endocrinology* 20 (2), S. 326–330 (1973).

Pakkala, J.J., Norris, D.R., and Newman, A.E.M. An Experimental Test of the Capture-Restraint Protocol for Estimating the Acute Stress Response. In: *Physiological and Biochemical Zoology* 86, S. 279–284 (2013).

Rintamäki, H., Hissa, R., Etches, R. J., Scanes, C. G., Balthazart, J., and Saarela, S. Seasonal changes in some plasma hormones in pigeons: diurnal variation under natural photoperiods with constant or seasonally changing ambient temperature. *Comparative biochemistry and physiology. A, Comparative physiology* 84, S. 33–38 (1986).

Shanahan, M., Bingman, V.P., Shimizu, T., Wild, M., Güntürkün, O. Large-scale network organization in the avian forebrain: A connectivity matrix and theoretical analysis. *Front Comput Neurosci* 7:89 (2013).

Van Ruijssevelt, L., Hamaide, J., Van Gorp, M. T., Verhoye, M. & Van Der Linden, A. Auditory evoked BOLD responses in awake compared to lightly anaesthetized zebra finches. *Sci. Rep.* 7, 1–12 (2017).

Van Ruijssevelt, L., Chen, Y., von Eugen, K., Hamaide, J., De Groof, G., Verhoye, M., Güntürkün, O., Woolley, S.C., Van der Linden, A. fMRI Reveals a Novel Region for Evaluating Acoustic Information for Mate Choice in a Female Songbird. *Curr Biol.* 28(5):711-721 (2018).

Westerhof, I., Mol, J.A., Van den Brom, W. E., Lumeij, J. T., Rijnberk, A. Diurnal Rhythms of Plasma Corticosterone Concentrations in Racing Pigeons (*Columba livia domestica*) Exposed to Different Light Regimens, and the Influence of Frequent Blood Sampling. In: *Avian Diseases* 38 (3), S. 428–434 (1994)

REVIEWERS' COMMENTS:

Reviewer #3 (Remarks to the Author):

I thank the authors for the extensive clarifications and modifications to the manuscript, and have no further objections against publication.